# Impaired Iron Homeostasis and Haematopoiesis Impacts Inflammation in the Ageing Process in Down Syndrome Dementia

**DOI:** 10.3390/jcm10132909

**Published:** 2021-06-29

**Authors:** Ruma Raha-Chowdhury, Animesh Alexander Raha, James Henderson, Seyedeh Deniz Ghaffari, Monika Grigorova, Jessica Beresford-Webb, Kieren Allinson, Subhojit Chakraborty, Anthony Holland, Shahid H. Zaman

**Affiliations:** 1Cambridge Intellectual & Developmental Disabilities Research Group, Department of Psychiatry, University of Cambridge, Cambridge CB2 8AH, UK; mg927@medschl.cam.ac.uk (M.G.); jb2192@medschl.cam.ac.uk (J.B.-W.); ajh1008@medschl.cam.ac.uk (A.H.); shz10@medschl.cam.ac.uk (S.H.Z.); 2John van Geest Centre for Brain Repair, Department of Clinical Neuroscience, University of Cambridge, Cambridge CB2 8AH, UK; alex.raha@icloud.com (A.A.R.); jh2014@cam.ac.uk (J.H.); sdg46@cam.ac.uk (S.D.G.); chaks210@aol.com (S.C.); 3Cambridgeshire and Peterborough Foundation NHS Trust, Cambridge CB21 5EF, UK; 4Clinical Pathology, Addenbrookes Hospital, Cambridge University Hospitals NHS Foundation Trust, Cambridge CB21 5EF, UK; kieren.allinson@addenbrookes.nhs.uk

**Keywords:** Alzheimer’s disease, accelerating ageing, down syndrome dementia, erythroid-megakaryocyte, iron homeostasis, trisomy 21, ferritin, hepcidin, iron, RUNX1, S100β, OLIG2, choroid plexus

## Abstract

Down syndrome (DS) subjects are more likely to develop the clinical features of Alzheimer’s disease (AD) very early in the disease process due to the additional impact of neuroinflammation and because of activation of innate immunity. Many factors involved in the neuropathology of AD in DS, including epigenetic factors, innate immunity and impaired haematopoiesis, contribute significantly towards the pathophysiology and the enhanced ageing processes seen in DS and as a consequence of the triplication of genes *RUNX1*, *S100**β* and *OLIG2*, together with the influence of proteins that collectively protect from cellular defects and inflammation, which include hepcidin, ferritin, IL-6 and TREM2. This study is aimed at determining whether genetic variants and inflammatory proteins are involved in haematopoiesis and cellular processes in DS compared with age-matched control participants, particularly with respect to neuroinflammation and accelerated ageing. Serum protein levels from DS, AD and control participants were measured by enzyme-linked immunosorbent assay (ELISA). Blood smears and post-mortem brain samples from AD and DS subjects were analysed by immunohistochemistry. RUNX1 mRNA expression was analysed by RT-PCR and in situ hybridisation in mouse tissues. Our results suggest that hepcidin, S100β and TREM2 play a critical role in survival and proliferation of glial cells through a common shared pathway. Blood smear analysis showed the presence of RUNX1 in megakaryocytes and platelets, implying participation in myeloid cell development. In contrast, hepcidin was expressed in erythrocytes and in platelets, suggesting a means of possible entry into the brain parenchyma via the choroid plexus (CP). The gene product of RUNX1 and hepcidin both play a critical role in haematopoiesis in DS. We propose that soluble TREM2, S100β and hepcidin can migrate from the periphery via the CP, modulate the blood–brain immune axis in DS and could form an important and hitherto neglected avenue for possible therapeutic interventions to reduce plaque formation.

## 1. Introduction

Down syndrome (DS) is the leading genetic cause of intellectual disabilities in live-born infants and is caused by the presence of an extra copy of chromosome 21 (Hsa21) [1,2,3]. People with DS have characteristic phenotypes including growth retardation, cognitive impairments, premature ageing, abnormal haematopoiesis, various forms of leukaemia and immune disorders [1,2,3]. There are many factors likely to be involved in the development and progression of the neuropathology of Alzheimer’s disease (AD) and in the enhanced ageing processes in DS, including epigenetic factors, innate immunity and impaired haematopoiesis [3,4,5,6]. The mechanism of the early onset of AD pathology in DS is not fully understood, but the over-expression of certain genes on Hsa21, particularly in the Down syndrome critical region 1 (DSCR1) that also includes the amyloid precursor protein (*APP*), *Cu/Zn superoxide dismutase* (*SOD1*), *RUNX1* (a transcription factor) [7] and *S100β,* play crucial roles [8,9]. Studies have demonstrated that dosage-dependent increases in Hsa21 gene expression do occur in trisomy 21 (T21) [10,11]. The many triplicated genes on Hsa21, particularly SOD1 and S100β, have been associated with oxidative stress, and their over-expression may contribute to a genetic imbalance in DS, and potentiate the development of reactive oxygen species (ROS) [12,13,14]. However, it is unclear whether the entire transcriptome is disrupted, or whether there is a more differential change in the expression of chromosome 21 genes [15,16]. It has been hypothesised that gene expression changes in Hsa21 are likely to affect the expression of genes located on other chromosomes through the modulation of transcription factors, immune regulation and other factors [17,18]. It is proposed that, in addition to primary gene dosage effects, secondary (downstream) effects on disomic genes are likely to have a major role in the development of the wider phenotype in the people with aneuploidies in general and DS in particular [9,10].

Triosomy-21 is associated with enhanced megakaryocytic and erythroid precursor expansion, and gene-encoded transcription factors *RUNX1, ERG, OLIG1* and *OLIG2* influence haematopoietic fates in DS [19,20,21,22]. Mutations in the *GATA-1 and RUNX1**,* which encode transcription factors, are central to the normal development of the erythroid and megakaryocytic lineages and are found in cases of myeloid leukaemia in DS children, but the incidence decreases in adulthood [23,24].

A key concern in adult DS subjects is an increase in vulnerability to the development of AD, which typically has an age of onset between 40 and 60 years [16,25]. A link between DS and AD in terms of dementia and neuropathological features has been established [26,27,28,29]. All adults with DS over the age of 40 years display neuropathological hallmarks of AD, including senile plaques (SPs) and neurofibrillary tangles (NFTs) [29,30,31,32]. SPs are primarily composed of amyloid beta peptide (Aβ42), a highly fibrillogenic form of the peptide, produced via sequential cleavage of the amyloid precursor protein (APP) by beta- and gamma-secretases [33]. The triplication of chromosome 21 in DS leads to the increased production of APP in a dose-dependent manner and could increase the deposition of Aβ42 protein, leading to the development of NFTs, one of the pathophysiological features of AD [34]. Additionally, high Aβ42 accumulation is toxic which could enhance the formation of ROS and oxidative stress in neurons, leading to neuroinflammation and premature cell death [13,35,36].

Aβ plaque deposition is an early event in DS brain, and Aβ and iron storage protein ferritin have been shown to co-localise in the vascular amyloid deposits of plaque in post-mortem AD brains [14,37,38,39]. Iron is involved in many fundamental biological processes in the brain, including oxygen transportation, DNA synthesis, mitochondrial respiration, myelin synthesis and for key enzymes involved in biosynthesis of, for example, neurotransmitters [40,41]. Many proteins are involved in iron homeostasis, including ferritin, hepcidin, transferrin and transferrin receptors, ferroportin (FPN) and divalent metal protein 1 (DMT1) [38,42,43,44]. Ferritin is an iron storage protein, and is often accompanied by abnormal iron homeostasis in inflammation that leads to systemic hypoferritinaemia [45]. Hepcidin is a member of the defensin family of antimicrobial cationic peptides that plays a significant role in host defence and innate immunity on account of its broad antibacterial and antifungal properties [46,47]. Hepcidin synthesis by hepatocytes is stimulated when body iron levels increase during systemic inflammation and is usually suppressed by hypoxia and increased erythroid activity [48,49]. Hepcidin is distributed in the circulation and extracellular fluid of cells that express the hepcidin receptor, ferroportin (FPN) [50]. This ferrous iron permease is the only known iron exporter expressed by mammalian cells, and hepcidin inhibits iron efflux by binding to the FPN on the cell surface and triggering the internalisation and lysosomal degradation of the complex [51].

Iron trafficking in the brain is less well-understood than systemic iron metabolism, although proteins implicated in iron transport and storage are common to both systems [52]. Hepcidin and its receptor ferroportin’s immunoreactivity are evident in neurons and glial cells, where it was suggested that both the proteins could play a role in iron homeostasis in the central nervous system [53]. It was reported that APP may bind to ferroportin to facilitate neuronal iron export, and that disturbance in this process may be implicated in AD neuropathology [54].

Inflammation has long been suspected to be a major player in the aetiology of AD dementia (ADD) in DS, and dysfunction of APP processing is believed to be the key upstream factor in neuroinflammation and for activation of innate immunity, with both being considered early events in the genesis of ADD [55]. In DS and in late-onset AD (LOAD), neuroinflammation has been linked to both the exacerbation of SP and NFT pathology, as well as the clearance of Aβ from amyloid plaques [56,57,58]. The main cell types involved in the brain neuroinflammatory responses are microglia, astrocytes and to a lesser extent, the peripheral and meningeal macrophages [39,55,59]. Microglia when activated release a cocktail of inflammatory cytokines and chemokines [60,61]. These are often triggered in a subcellular compartment known as the inflammosome, where IL-6 and IL-1β are the main pathological mediators causing cytokine release [39,61].

Besides iron levels in the circulation, inflammation is yet another factor regulating hepcidin transcription in the hepatocytes. Inflammatory cytokines, mainly in the form of IL-6, are released during inflammation and induce hepcidin expression in the hepatocytes via the JAK/STAT3 signalling pathway, leading to phosphorylation of STAT3, its translocation to the nucleus and activation of the hepcidin gene [62]. The identification of a novel variant in the gene encoding the triggering receptor expressed on myeloid cells 2 (TREM2) has also refocused attention on inflammation as a major contributing factor in AD. Variants in TREM2 (rs75932628, substitution of T to C, changes in R47H mutation) significantly increase the risk of developing late-onset AD, by three-fold [29,63,64,65,66]. TREM2 is a pattern recognition receptor, expressed on macrophages and microglia. It has the function to detect lipid and Aβ that stimulate phagocytosis and induce an innate immune response [65,67].

The aim of our study was to explore the possible role of the proteins triplicated in T21 and how these proteins are involved in transport of amyloid protein, iron metabolism and influence neuroinflammation and cellular damages seen in DS. We evaluated hepcidin, ferritin, SOD1, S100β, IL-6 and TREM2 expression in serum and post-mortem brain samples to determine the crosstalk between inflammation and iron dysregulation in normal ageing and in DS pathology and to ascertain how these changes in neuroinflammation could play a role in the acceleration of ageing in DS. To explore progression of abnormalities in iron homeostasis in DS, we also investigated blood smears from DS and age-matched controls to determine the role of RUNX1 and hepcidin in the development of the erythroid and megakaryocytic lineages.

## 2. Materials and Methods

### 2.1. Ethics and Participants

The National Research Ethics Committee of the East of England granted approval for the study. The Cambridge Health Authorities Joint Ethics Committee granted ethical approval for the use of human brain tissue and serum samples (Project ref no: REC:15/WM/0379). Written consent was obtained from controls and from adult DS participants who had the capacity to consent. Verbal assent was obtained from participants with DS, who lacked the capacity for consent and written assent, and was provided by an appointed consultee, in accordance with the English and Welsh Mental Capacity Act of the UK (2005). Information on older controls (*n* = 25), younger controls (*n* = 25) and DS participants (*n* = 47) has been reported in previous publications [29,66].

### 2.2. Assessment of Dementia Status

Dementia status assessment was undertaken as described previously using the CAMDEX-DS informant interview and the CAMCOG-DS neuropsychological assessment [68].

### 2.3. Blood and Serum Collection

Whole blood and serum samples were collected from human controls (*n* = 50, aged between 30 and 85 years), with intact cognitive functioning and devoid of neurological and mental illness, and from 47 participants with DS (aged between 32 and 70 years) and assessed at our Research Centre. All blood samples were collected for serum in BD vacutainer SST advance tubes (containing inert gel barrier and clot activator coating). Serum and plasma were separated immediately by centrifugation at 2465× *g* for 6 min at 4 °C, aliquoted, and stored at −80 °C until analysis. Biochemical and haematological profiles including serum iron were analysed by pathology laboratories at Addenbrooke’s Hospital, Cambridge University NHS Foundation Trust, Cambridge, UK.

### 2.4. Human Post-Mortem Brain Sections

Human post-mortem brain tissues from controls (mean age 60 ± 15 years) and from people with DS (mean age 60 ± 15 years) (N = 10 in each group) were provided by the Cambridge Brain Bank (Table 1). The Cambridge Health Authorities Joint Ethics Committee granted ethical approval for use of human brain tissues and serum samples (Project ref no: REC:15/WM/0379). All methods have been described in our previous publications [29,39,65,66].

### 2.5. Blood Slide Preparation

A drop of blood placed on a slide was spread as described previously [29,61]. The slide was air-dried and fixed in 100% methanol and stored at 4 °C. Further investigations, including immunohistochemistry, were performed on stored fixed slides, as described later. Bone marrow slides from DS participants were fixed with 100% methanol and stored at 4 °C.

#### 2.5.1. Animals

Three-month-old C57/Bl6 mice were purchased from Charles River and bred for the experiments at the University of Cambridge Bioscience facility (UBSS, at John van Geest Centre for Brain repair animal facility). All experiments were performed according to the “Animal (Scientific Procedures) Act 1986” and the “Guidance of the Operation of ASPA 2014” and were approved by the Ethical Committee of the UK Home Office.

#### 2.5.2. Mouse Tissue Preparation

All animals were housed under standard conditions (12 h light–dark cycle, 20 °C ambient temperature) with free access to food and water. Animals undergoing perfusion were anaesthetised with rising concentrations of CO_2_ followed by terminal injection of Euthatal (0.3 mL per 100 g of body weight). Collection of fresh tissues was performed after raising the concentration of CO_2_ followed by cervical dislocation. For histochemical analyses, animals were anaesthetised with pentobarbitone and flash-perfused transcardially with 0.9% saline, followed by 4% (*v*/*v*) paraformaldehyde (PFA) in 0.1 M phosphate buffer saline (PBS, pH 7.4). Brains were removed, post-fixed for 4 h in the same fixative, and then cryoprotected with 30% sucrose in 0.1 M PBS. Brains and other tissues (spleen and duodenum) were sectioned by microtome as described previously [38]. For in situ hybridisation, unfixed tissues were carefully dissected from C57/Bl6 mice from various brain regions and different organs, including spleen, lung, duodenum and embryos (embryonic day 16, ED16), and snap-frozen in dry ice until they were analysed by in situ hybridisation.

#### 2.5.3. RNA Isolation and RT-PCR

Total RNA was extracted from ED16 brain, ED16 whole body and adult mouse tissues. Approximately 50 mg of tissue was placed in 1 mL of TRIzol reagent (Invitrogen, Waltham, MA, USA) following the manufacturer’s instructions and then purified using a RNeasy Mini kit (Qiagen, Hilden, Germany). RNA was treated with DNase I and cDNA synthesised from 2 g of the treated RNA using a SuperScript III reverse transcriptase kit (Invitrogen) with random hexamer primers. Real-time PCR was performed using iQ Supermix SYBR Green 2 by Bio-Rad on the Bio-Rad iCycler (www.bio-rad.com (accessed on 15 January 2021)) according to the manufacturer’s protocols. PCR reactions were performed in triplicate. The primers used for amplification were RUNX1 (Forward Primer: 5′CAGGCAAGATGAGCGAGGCGTTG3′ and Reverse primer: 5′CAACCTGGTTCTTCATGGCTGC3′, product size 320 bp), using a program of 95 °C for 3 min, 30 cycles of 95 °C for 30 s, 60 °C for 30 s and 72 °C for 1 min. The GAPDH housekeeping gene was chosen as the reference gene. GAPDH (Forward Primer: 5′CGGAGTCAACGGATTTGGTCGTAT3′, Reverse primer: AGCCTTCTCCATGGTGGTGAAGAC, (product size 500 bp).

#### 2.5.4. In Situ Hybridisation (ISH)

Brain tissues and other organs were prepared for in situ hybridisation and probed as described previously [61]. To synthesise DIG-labelled RNA probes, the target RUNX1 cDNA was amplified by PCR using primers designed on the basis of the mouse RUNX1 cDNA sequence used in RT-PCR. The primers used for DIG labelling were: RUNX1-insF: 5′TAATACGACTCACTATAG GCAGGCAAGATGAGCGAGGCGTTG3′, and RUNX1-insR: 5′ATTTAGGTGACACTATAGAGCAACCTGGTTCTTCATGGCTGC3′, sequenced and homology checked by BLAST search (NCBI database). In vitro transcription reactions were performed using dig-UTP RNA labelling mix and SP6 or T7 RNA polymerase (Roche, Mannheim, Germany). The sense and anti-sense DIG-labelled probe was prepared to assess hybridisation stringency, and freshly frozen mouse tissues from liver, brain, duodenum and embryonic body (ED16).

This method has been described in detail in our previous publications [53,61]. Briefly, *ISH* with DIG-labelled probe was carried out on liver, brain and other embryonic tissue sections fixed in 4% PFA and permeabilised in 0.1 M PBS with 0.5% Triton X-100, and acetylated by incubation in triethanolamine solution. Pre-hybridisation was performed for 3 h at 62 °C and then hybridised with DIG-labelled probes (100 ng/mL) in the same buffer overnight at 62 °C. Stringent washing was performed in 0.2× SSC for 1 h at 62 °C. For the detection of DIG-labelled hybrids, the slides were equilibrated in maleic acid buffer (MAB) at room temperature (RT) with 1% blocking reagent made in MAB (blocking buffer), and then for 1 h with alkaline phosphatase-conjugated anti-DIG antibodies (Roche, Mannheim, Germany) diluted 1:5000 in blocking buffer. The slides were washed and incubated overnight in colour development buffer (2.4 mg levamisole (Sigma, Kawasaki, Japan), 45 μL 4-nitroblue tetrazolium (Sigma) and 35 μL 5-bromo-4-chloro-3-indolyl-phosphate (Sigma)). The reaction was stopped in neutralizing buffer (0.01 M Trizma base and 0.001 M EDTA, pH 8) and sections were mounted in PBS–glycerol and a coverslip was applied over the section on the slide. Non-specific binding was analysed using sense probes.

#### 2.5.5. SDS-PAGE Gel and Western Blotting

Protein samples were prepared from human serum (DS and age-matched controls, *n* = 20). Twenty μg protein samples were separated on 4–12% Nu-PAGE Bis-tris (Bis (2-hydroxyethyl)-amino-tris (hydroxymethyl)-methane) gradient gels and transferred to 0.2 μm pore size PVDF for Aβ42, hepcidin and S100β, and 0.45 μm pore size PVDF membranes for SOD1, FTL, FPN, TREM2 and albumin using the NuPAGE electrophoresis system (Invitrogen). Membranes were incubated with different antibodies, i.e., anti-β amyloid (Covance, SIG 39320), SOD1 (Abcam, Cambridge, UK, ab252426), S100β (Abcam, ab218514), anti-ferritin light chain (FTL, Abcam, ab69090), anti-FPN (Abcam, ab85370), anti-hepcidin 25 (Abcam, ab30760), anti-TREM2 (Abcam, ab86491) and anti-albumin (Abcam, ab10241) antibody, in blocking buffer for 24 h at 4 °C, and then washed three times with 0.1 M Tris saline buffer containing 1% Tween 20 (TBST), followed by incubation for 1 h at RT with HRP-conjugated secondary antibodies, anti-rabbit IgG (1:3000, DAKO) or anti-mouse IgG (1:3000; DAKO) antibodies. Binding was detected with ECL Plus chemiluminescence reagents.

### 2.6. Solid Phase Enzyme-Linked Immunosorbent Assay (ELISA)

The human serum ferritin level was measured with the human ferritin ELISA kit (Abcam, catalogue number ab108698) and serum hepcidin (human hepcidin Quantikine ELISA kit, catalogue number DHP250), IL-6 (human IL-6 Quantikine ELISA kit, R&D, catalogue number S6050) and TREM2 (Thermo Fisher, catalogue number EH464RB) according to the manufacturer’s instructions. Briefly, for the detection of hepcidin, 96-well culture plates were incubated overnight with monoclonal antibody (R&D, MAB83071) or anti-hepcidin capture antibody (1 μg/mL) (R&D system, catalogue number 842127). The following day, the plates were washed three times with washing buffer (0.05% Tween 20 in 0.1 M phosphate buffer saline (PBS) pH 7.4) and blocked in blocking solution (1% BSA and 0.05% Tween 20 in 0.1 M PBS) for 2 h, at RT. After blocking, the plates were washed three times with washing buffer and loaded with 10 μL plasma into 90 μL blocking solution and incubated for 4 h, at RT. All experiments were performed in quadruplicate unless otherwise specified. A recombinant human protein (R&D, catalogue number 842129) was diluted in assay buffer in a two-fold serial dilution and used for the standard curve, with a concentration range of 1000, 500, 250, 125, 62, 31, 15 and 0 pg/mL. After 4 h of incubation, the samples were removed, and the plates were washed three times for 5 min with washing buffer before incubation for 2 h at RT for detection with a biotinylated rabbit polyclonal anti-human hepcidin (Abcam, ab30760) antibody (1 μg/mL), diluted in blocking buffer. After three further washing steps, the plates were incubated with anti-rabbit HRP-conjugated secondary antibody (1:4000) for 1 h followed by three washes. Then, 100 µL of 1-Step ULTRA tetramethylbenzidine (TMB-ELISA, ThermoScientific) was added for ~30 min at RT. Finally, 100 µL of 2 M H_2_SO_4_ was added to quench the reaction. Colorimetric quantification was performed with an Infinite M200 plate reader (Tecan, Männedorf, Switzerland) at 450/540 nm.

### 2.7. Antibodies

A polyclonal rabbit anti-hepcidin 25 (Abcam, ab30760), recognising a 2.8 kDa protein [53], monoclonal antibody (MAB) human anti-ferritin light chain (Abcam, ab69090), human anti-ferritin heavy chain (Abcam, ab65080), anti-DMT1 (Abcam, ab123085), anti-FPN (Abcam, ab85370), anti-CD42b (Abcam, ab104704), anti-rabbit polyclonal glycophorin (Abcam, ab196568), anti-GFAP (Abcam, ab48050), anti-CD68 (Sigma-Aldrich, AMAB98073), MAB anti-CD11b (Thermo Fisher, Waltham, MA, USA, mAbM1/70), GFAP (Abcam, ab48050), MAB anti-Iba1 (Thermo Fisher, MAB M1/70), polyclonal anti-Iba1 (Wako, Fujifilm, Tokyo, Japan, catalogue number 019-19741), MAB anti-IL-6 (Thermo Fisher, catalog number M620), MAB IL-1β (Thermo Fisher, ILB1-H67), MAB anti-β amyloid 42 (Covance, Princeton, NJ, USA, SIG 39320), anti-phospho-tau (AT8, Thermo Fisher, MN1020), SOD1 (Abcam, ab252426), S100β (Abcam, ab218514), RUNX1 (Sigma-Aldrich, HPA004176) and OLIG2 (Santa Cruz Biotechnology, Dallas, TX, USA, sc-365644) were used for IHC or Western blotting. The following secondary antibodies were used: biotinylated goat anti-rabbit-Ig and biotinylated horse anti-mouse (both from Vector Laboratories, 1:250 for IHC), Alexa Fluor 568-labelled donkey anti-mouse-Ig, Alexa Fluor 488-labelled donkey anti-rabbit-Ig and Alexa Fluor 568-labelled donkey anti-goat-Ig (all from Invitrogen, 1:1000 for immunofluoresence).

### 2.8. Immunohistochemistry (IHC)

PFA fixed tissues were first quenched with 5% hydrogen peroxide and 20% methanol in 0.01 M PBS for 30 min at RT, followed by three rinses for 10 min in 0.01 M PBS. Non-specific binding sites were blocked using blocking buffer (0.1 M PBS, 0.3% Triton-X100 and 10% normal goat serum for polyclonal antibodies or 10% normal horse serum for monoclonal antibodies) for 1 h, at RT. Tissue sections were incubated overnight with the primary antibody diluted in blocking buffer. Binding of the primary antibody was detected using a biotinylated secondary antibody followed by an avidin-biotin complex conjugated to peroxidase (Elite standard kit SK6100, Vector Laboratories, Inc., Burlingame, CA, USA) and DAB substrate (ABC substrate SK-4200, Vector Laboratories).

### 2.9. Immunofluorescence (IF)

Brain and other sections were blocked using blocking buffer (0.1 M PBS, 0.3% Triton X100, 10% normal donkey serum) for 1 h, at RT, then incubated overnight at 4 °C with primary antibody diluted in blocking buffer. Alexa Fluor-conjugated secondary antibodies were used for detection and samples were counterstained with 4′6-diamidino-2-phenylindole (DAPI, Sigma). Sections were then mounted on glass slides with coverslips using Fluoro Save (Calbiochem, Merck KGaA, Darmstadt, Germany).

### 2.10. Microscopy

Bright-field images were taken and quantified using Lucia imaging software and a Leica FW4000 upright microscope equipped with a SPOT digital camera. Fluorescence images were obtained using a Leica DM6000 wide-field fluorescence microscope equipped with a Leica FX350 camera with 20× and 40× objectives. Images were taken through several z-sections and de-convolved using Leica software. A Leica TCS SP2 confocal laser-scanning microscope was used with 40× and 63× objectives to acquire high-resolution images.

### 2.11. Statistical Methods

Data were analysed by paired Student’s *t*-test (two-tailed) for two group comparison, or by ANOVA for multiple comparison testing. Values in the figures are expressed as mean ± SEM. A one-way ANOVA was used for comparison of data among control, AD and DS, and was conducted with IBM-SPSS statistics 19 software. Significance was analysed using GraphPad, and *p*-values ≤ 0.001 were considered significant and are indicated in the corresponding figures and figure legends.

## 3. Results

### 3.1. Characterisation of Senile Plaques in DS Brain

Senile plaque (SP) formation is the key neuropathological hallmark in AD and DS, and SPs are primarily composed of amyloid beta peptide (Aβ42). To characterise the SP formation, the brain sections from superior frontal gyrus (SFG), mid temporal gyrus (MTG) and hippocampus from DS and control brains (*n* = 10 in each group, average age between 42 and 70 years) were analysed by IF and imaged with confocal microscopy, using antibodies specific to Aβ42 (6E10), phospho-tau (pTau, with AT8 antibody), hepcidin and TREM2. In a younger (42 years old) DS brain, we stained SFG sections with Aβ42 antibody and found abundant cotton wool appearance of SPs throughout the cortex (Figure 1A, white arrows highlight the selected area of plaque formation). Most of the plaques were formed from a cluster of small swollen neurites surrounded by neuropil without an amyloid core, whereas in the periphery, some other mature SPs were surrounded by a halo of Aβ42 (Figure 1A, short arrow highlights the selected areas of plaque formation). To compare with non-DS AD brain, a section was stained with Aβ42 and pTau and analysed by confocal microscope. We observed a mature SP that contained many pTau-positive dystrophic neurites in the centre, with distinct Aβ42-positive astrocytes in the surrounding area (Figure 1B). For further analysis, another DS brain section from MTG was stained with pTau and TREM2, a well-known inflammatory marker. NFTs were visible in the core of the plaques with some co-localisation with TREM2 in the SP and close by (Figure 1C). A confocal image of DS MTG showed pTau-positive NFT visible very close to a blood vessel (Figure 1D). As most of the SP is composed of astrocytes, another DS SFG sample was stained with an astrocytic marker (GFAP) and imaged using confocal microscopy. The SP was composed of GFAP-positive astrocytes and it was surrounding the endothelial layer of blood vessels (Figure 1E). To compare with controls, one SFG from a control brain section was stained with Aβ40 and hepcidin. Amyloid proteins were found to be evenly distributed in the neuronal cell body, axons and dendrites, whereas hepcidin was present in the cell bodies only (Figure 1F). These findings indicate that not only neurons but also astrocytes contribute a great deal to plaque formation.

### 3.2. Hepcidin and S100β-Positive Activated Damaged Astrocytes Seen around the Wall of Blood Vessels in DS

Hepcidin is a key iron regulatory protein and is involved in iron transport from the brain parenchyma via astrocytes. To evaluate the expression of hepcidin in the astrocytes, sections from the cortex close to the blood vessels were analysed. In DS brain, GFAP-positive activated astrocytes were present around the SP and surrounding the blood vessels that also co-localised with hepcidin in the endothelial cells (Figure 2A–C, indicated with an arrow). In the DS brain, particularly in the SP, a dense core of GFAP-positive astrocytes were visible in a very organised form, where hepcidins located in the vesicles and co-localised in the astrocytes (Figure 2D–F). In a control brain section, normal astrocytes with fine processes were visible and co-localised with hepcidin in the cell bodies (Figure 2G). A large number of small vesicles (exosomes) carrying hepcidin were present in the SP of the DS cortex and its surroundings (Figure 2H). Hepcidin expression was visible in the periphery of blood vessels (in the endothelial cells) and co-localised with the GFAP-positive astrocytes in the DS brain (Figure 2I). Very strong co-localisation of hepcidin was noticed in layers I and II of the cortices (Figure 2I).

S100β promotes development and maturation in the mammalian brain, and its role in the cascade of glial changes associated with neuroinflammation is well-known [69]. Therefore, we stained brain sections from SFG of control and DS subjects with S100β and inflammatory protein TREM2 and analysed them by confocal microscopy. In control brain, normal astrocytes with thin filaments of processes of end-feet were present and surrounded the blood vessels, and were positive for S100β (Figure 2J). TREM2 were present in the astrocyte’s end feet carrying soluble TREM2 and co-localising in the cell bodies (Figure 2J–L). Some of the astrocyte’s filaments were directly connected to the endothelial lining of blood vessels (Figure 2J–L). However, in DS brain, S100β-positive activated astrocytes were disorganised and aggregated around the blood vessel walls, indicating severe endothelial cell damage of blood vessels (Figure 2M–O). This finding suggests that S100β is important in astrocytes and cellular changes seen in ageing-associated DS pathology, particularly of gliosis.

### 3.3. Hepcidin and Ferroportin Expression in the Oligodendrocytes of White Matter Tract

Hepcidin and ferroportin (FPN) could be involved in glial activities not only in astrocytes but also in oligodendrocytes to transport amyloid and other proteins through the white matter tract (WMT). To assess this aspect, we extended our investigation of hepcidin and FPN in white matter (WM) of controls and DS brains. Brain sections from SFG, MTG and hippocampus (HP) from DS and controls were then analysed by IHC using antibodies specific to Aβ42 (6E10), hepcidin and FPN, and using DAB stain. In the SFG of DS brain, abundant Aβ42-positive SPs were observed in the neocortex (Figure 3A, black arrows highlight the selected area of plaques formed) and hepcidin was also visible in the SPs (Figure 3B). In control brains (cortex), abundant hepcidin was seen in the neurons and in the neuropils (Figure 3C), whereas in the DS cortex, hepcidin staining was faint, present only in the neurons and glial cells (Figure 3D). In DS brain, hepcidin was present in the granule cells of the dentate gyrus (DG) (Figure 3E), and in the basal ganglia (BG) as punctate accumulation of vesicular hepcidin in the cell bodies and around the blood vessels (Figure 3F). In the WMT, typical hepcidin staining was visible as normally appearing in the control brain (Figure 3G), whereas a very granular appearance was visible in the cell bodies of oligodendrocytes in the DS brain (Figure 3H). FPN expresses in the pyramidal neurons of HP in control brain (Figure 3I), however very faint staining was seen in the pyramidal neurons in the DS brain (Figure 3J). Similarly, in DS brain, FPN expression was very faint and irregular in the WMT (Figure 3K).

### 3.4. DMT1, FPN and Hepcidin Regulate Endothelial Transport and SOD1, Modulating Redox Balance in the DS Brain

The function of iron (import/export) was studied in gut mucosa, mainly by duodenal membrane binding protein divalent metal protein 1 (DMT1) and FPN [42,43,44]. How these proteins regulate iron import and export in the brain parenchyma is still unclear. For validation of DMT1 and FPN in the gut mucosa and other endothelial cells, tissue sections from mouse duodenum and choroid plexus (CP) were stained by IFC using anti-DMT1 and anti-FPN antibodies. The most distinct feature of the small intestine is the mucosal lining, with brush border villi and crypts, lined with columnar cells. DMT1 staining was seen in the columnar cells within the crypts, particularly in the apical surface (Figure 4A) and FPN at the basolateral site and in the central core of lamina propria containing blood vessels (Figure 4B,C). These proteins are involved in absorbing iron from the luminal surface and transporting to the blood at the abluminal site via FPN (Figure 4A–C).

As we previously reported, the CP is damaged in DS and AD brains, and CP is a conduit between the peripheral circulation and central nervous system via the cerebrospinal fluid [61,66]. We extended our investigation to CP sections from mouse brain (Figure 4D–F) and human brain, and stained with DMT1, FPN and hepcidin (Figure 4G–I). DMT1 expressed in the monolayer of the epithelial cells of CP on the luminal sites, whereas FPN was found in the abluminal site of the CP with some co-localisation (Figure 4D–F). Similarly, in the human brain, hepcidin was visible in the epithelial cells of CP, whereas FPN was seen in the macrophages very close to CP (Figure 4G–I). These findings suggest that hepcidin might be transported via exosomes or other small vesicles and FPN via macrophages to the brain parenchyma. Further, hepcidin and FPN are both visible in the oligodendrocytes located in the corpus callosum (CC), suggesting that iron transport may take place in the CC (Figure 4J–L). Both proteins are transported via blood vessels (Figure 4M,N). Other brain sections from cortices close to SP were stained with SOD1 antibody, and very strong SOD1 protein staining was seen in the plaques and in the peripheral neurons (Figure 4O), indicating a mechanism of ROS scavenging in plaques of DS brain. To evaluate the role of SOD1 in the neurons and particularly in mitochondrial metabolism, we have to further explore it at a cellular level and in the serum.

### 3.5. Inflammatory Changes in the DS Brain Were Identified by TREM2, IL-1β and IL-6

Hepcidin and TREM2 could be involved in microglial activity and in the amyloid clearance process [38,53]. To assess this aspect, we extended our investigation to include hepcidin and TREM2 expression in microglia in control and DS brain samples (*n* = 10 from each group, Table 1), with a particular focus on identifying TREM2 expression in the Iba1-positive microglia. The brain sections from DS and control subjects were stained with microglia marker (Iba1) and TREM2. In control brains, Iba1-positive ramified microglia were visible in the cortex, whereas TREM2 was noticed in the neurons close to the blood vessels (Figure 5A). There was expression of TREM2 in some cells but not in Iba1-positive microglia (Figure 5A). Similarly, in the DS brain, activated microglia were visible around the SP with limited co-localisation with TREM2 in the microglia (Figure 5B). In sections from the HP and DG of another DS brain where most of the granule cells were damaged, when stained with Iba1 and TREM2, Iba1-positive activated microglia were present close to the neurons, whereas TREM2 expression was visible only in the damaged DG granule cells with limited co-localisation (Figure 5C,D). These findings indicate that there are different populations of microglia sub-types present in the human brain. We further investigated white matter (WM) close to the blood vessels (BV) in the DS brain and found large numbers of Iba1-positive microglia, close to the blood vessels, that could be involved in the clearing process of damaged oligodendrocytes, and some TREM2-positive macrophages also seem to have entered from blood vessels (Figure 5E). For another DS brain including BV, when stained with hepcidin and IL-6, the staining pattern suggested that both proteins might be entering from the blood vessels (Figure 5F).

To evaluate the effects of inflammatory changes in different glial cells, brain sections from controls and DS (cortex) subjects were stained with TREM2 and pro-inflammatory marker IL-1β. In control brain sections, TREM2 was visible in the neurons and in other cell types, and only very faint IL-1β expression was seen with limited co-localisation in the cell bodies (Figure 5G–I). In DS brain, however, very strong TREM2 expression, particularly in the neuronal cells, was observed, whereas IL-1β was noticed in the microglia, close to the blood vessels and with some co-localisation noted (Figure 5J–L). Similarly, control and DS brain sections, when stained with hepcidin and IL-6, showed that both the proteins were present in the cells close to the blood vessels and with some co-localisation (Figure 5M). In contrast, in the DS brain, damaged neurons with a halo around the nucleus were positive for hepcidin, and IL-6 was visible in the microglia (Figure 5N,O). These findings indicate that inflammatory cytokines enter in the brain parenchyma through the blood vessels. TREM2 is an anti-inflammatory protein that did not co-localise with Iba1-positive activated microglia, supporting our previous findings [65].

### 3.6. OLIG2 Defect Impairs Myelination in DS Brain

We previously reported that WMT defects were seen in DS brain. Additionally, we have shown that in the rat brain, hepcidin and FPN were expressed in the white matter tract (WMT) of the corpus callosum (CC) and are involved in myelination [38,53,66]. OLIG2 promotes the formation and maturation of oligodendrocytes within the brain [70]. We investigated OLIG2 and myelin basic protein (MBP) expression in control and DS white matter (WM) in the cortex and in the basal ganglia (BG). In the control brain, MBP expression was seen in the myelinated sheath of oligodendrocytes, with OLIG2 localising in the cell bodies (Figure 6A–C), and in the DS brain, myelin sheaths appeared shrunken and damaged (Figure 6D–F). In some cases of DS brain, Aβ42-positive SPs were visible in the WM of the cortex (Figure 6G). In another DS brain section from BG, when stained with hepcidin and S100β, both proteins were seen in the WM. S100β, in particular, was present in the oligodendrocytes (Figure 6H,I), whereas hepcidin was seen in the wall of blood vessels (Figure 6I). In control brain, particularly in the BG, MBP and OLIG2 were positive in WMT (Figure 6J,K), whereas, in DS brain, Aβ42-positive SP were visible (Figure 6L). These findings indicated that triplication of OLIG2 may disturb the myelination process in DS.

### 3.7. Serum Hepcidin and IL-6 Could Be Involved in Host Defence Mechanism in DS Brain

To investigate the effect of trisomy-21-related proteins (Aβ42, SOD1, S100β) and iron regulatory proteins (FTL, FPN, hepcidin), as well as inflammatory marker TREM2, on serum levels, a cohort of DS subjects (*n* = 18) and age-matched controls (*n* = 18) was analysed by Western blotting. Aβ42 (4 kDA) and S100β (11 kDA) levels were higher in the DS subjects compared to the controls (Aβ42, *p* < 0.084, S100β *p* < 0.26, paired t-test, not significant) (Figure 7A and Appendix A). In contrast, SOD1 (18 kDA) level was found to be very high in DS serum (R^2^ = 0.64, *p* < 0.0001, Appendix A). However, FTL (19 kDA), FPN (69 kDA) and TREM2 (25 kDA) were significantly decreased in the DS serum (*p* < 0.0001, *p* < 0.003 and *p* < 0.002 respectively, Appendix A). By contrast, hepcidin was significantly higher in DS serum (*p* < 0.001) (Figure 7A, Appendix A).

Due to the large number of serum samples and the heterogeneity of different batches of blots generated by WB analysis, we decided to further analyse using ELISA, with reliable kits available for the tests. We investigated serum ferritin, hepcidin, TREM2 and IL6 levels in ‘young DS’ (YDS, *n* = 23, 30–45 years) and ‘older DS’ (ODS, *n* = 24, 46–70 years) participants and then compared them with age-matched controls (young, YC, and old, OC, *n* = 25 in each group) using sandwich ELISA. All samples were analysed on the same day, using the same standardised protocol to reduce the day-to-day variations. Serum iron was measured by clinical biochemistry, at the Addenbrooke’s Hospital of Cambridge University NHS Foundation Trust located in Cambridge, UK, as described in our previous publication [39]. Serum iron levels were significantly higher (*p* < 0.001) in young controls (YC, 25.07 ± 5.06 μmol/L) and in old controls (OC, 20.18 ± 7.2 μmol/L) compared to young DS (YDS, 19.11 ± 4.5 μmol/L) and old DS (ODS, 13.92 ± 7.4 μmol/L). ELISA data were evaluated (in four groups) by nonparametric statistical analysis (Kruskal–Wallis and post hoc with Mann–Whitney U test, R^2^ 0.49, *p* < 0.0001, Table 2).

Hepcidin levels were substantially higher in DS (YDS, mean ± SD: 376.36 ± 475.4 μg/L), and ODS, 241.02 ± 430.5 μg/L) compared to both young and old controls (YC, 23.89 ± 19.1 μg/L, and OC, 26.56 ± 16.1 μg/L, R^2^ 0.35, *p* < 0.0001, Figure 7B). All data were analysed using one-way ANOVA, which compared between groups, followed by Kruskal–Wallis or Bartlett’s tests, corrected (Table 2), and paired *t*-tests, between YC vs. YDS and YC vs. OC, as shown in Appendix A. Similarly, serum ferritin was highest in controls (YC, 216.9 ± 171.1 μg/L, and OC, 211.43 ± 45.5 μg/L, Appendix A), and lowest in the young DS (YDS, 122.56 ± 10.4 μg/L, compared to ODS, 139.83 ± 85.7 μg/L) (Figure 7C, one-way ANOVA, R^2^ 0.31, *p* < 0.0001, Table 2). Serum TREM2 levels were the highest in young controls (YC, 1190.91 ± 150.2 pg/mL, and OC, 796.26 ± 147.65 pg/mL) and the lowest in the old DS (YDS, 894.30 ± 201.52 pg/mL, and ODS, 562.79 ± 112.75 pg/mL, R^2^ 0.48, *p* < 0.0001) (Figure 7D, Table 2). Serum IL-6 was higher in DS (YDS, 244.29 ± 144.4 μg/L, and ODS, 343.14 ± 320.1 μg/L) compared to both controls (YC, 52.62 ± 21.01 μg/L, and OC, 76.55 ± 31.4 μg/L, R^2^ 0.63, *p* < 0.0001) (Figure 7E). Serum hepcidin and IL-6 levels were both highest in the old DS subjects (R^2^ 0.47, *p* < 0.0001, Figure 7F,G, Table 2). In contrast, serum TREM2 was higher in the controls compared to DS participants (R^2^ 0.77, *p* < 0.0001, Figure 7H, Table 2). These findings suggest that serum hepcidin and IL-6 could be involved in host defence mechanisms seen in neuroinflammation.

### 3.8. RUNX1 mRNA Expression Was Higher in the Liver, Bone Marrow and Duodenal Endothelial Cells, as Assessed by In Situ Hybridisation (Evaluated in Mouse)

The runt-related transcription factor RUNX1 gene is located in human chromosome 21 and is highly conserved in many species, including humans and mice. Mutations and translocations in the RUNX1 gene are frequently seen in acute myeloid leukaemia (AML), myelodysplastic syndromes (MDS) and other forms of haematological malignancies common among DS subjects [71]. Therefore, we validated the RUNX1 mRNA expression pattern by RT-PCR, in wild-type mouse (C57/bl6) tissues from ED16 body, brain (cortex), heart, muscles, duodenum, liver, bone marrow and peripheral leukocytes (Figure 7G). This analysis revealed low levels of mRNA in the brain, with a slightly higher signal observed in the heart, skeletal muscle, gut, liver and with the highest expression seen in the bone marrow and peripheral mononuclear leukocytes (PBMC) (Figure 7I).

To investigate differential distribution of mRNA in different organs, we extended our findings to cellular levels by in situ hybridisation (ISH) using a RUNX1 probe (320 base pairs) amplified by DIG labelling, and freshly frozen mouse tissues from liver, brain, duodenum and embryonic body (ED16) were analysed. Very high mRNA levels were visible in ED16 mouse brain, throughout the cortex and brain cavity (Figure 7J), but less in the adult brain, where they were only visible in the HP and DG and clearly in the brain (Figure 7J,K) and in the blood vessels (in the blood cells) (Figure 7L). However, higher mRNA expression was seen in the bone marrow, endothelial layer of the heart and duodenum (Figure 7M,N,P). The highest mRNA level was seen in the liver (Figure 7O). When another spleen section was hybridised with control probe (ferritin), strong mRNA was visible (Figure 7Q). Taken together, these in situ data suggest that RUNX1 is transcribed in the liver, bone marrow and endothelium of the blood vessels, and could be the likely explanation of the higher signal seen with RT-PCR.

We stained ED16 mouse brain with RUNX1 and hepcidin antibodies. Both proteins were observed in the embryonic brain, in the ED16 body and spinal cord (Figure 7R–T), with co-localisation in the epithelial cells of the placenta (Figure 7U). RUNX1 was distinctly visible in the embryonic liver (Figure 7V), endothelial cells of the gut and in the muscles (Figure 7W,X), and co-localised with hepcidin in the adult liver hepatocytes (Figure 7Y). These data suggests that RUNX1 protein highly expresses in the embryonic stages and in adult stages, particularly in those cells that have self-renewal properties.

### 3.9. RUNX1 and Hepcidin Were Observed in the Erythro-Megakaryocytes and Platelets, Suggesting a Myeloid Origin

RUNX1 affects crucial processes such as self-renewal of haematopoietic stem cells and differentiation of the myeloid and lymphoid cell lineages [24]. There is an increased risk of leukaemia in adults with DS, and a variety of haematological abnormalities, including abnormal erythrocytosis, macrocytosis and leukopenia are also seen [71]. Therefore, we analysed RUNX1 and hepcidin expression in blood smears from DS participants (*n* = 20, from different age groups between 30 and 65 years) and age-matched controls and then imaged them with confocal microscopy. DS and control blood smears were co-labelled with RUNX1 or hepcidin and glycophorin, an erythrocyte plasma membrane-marker. Both proteins were found to co-localise in the RBC membranes, suggesting that RUNX1 expresses on the surface of RBCs (Figure 8A, Table 3). Hepcidin was observed within the cell bodies of RBCs and mononucleated cells (MNCs) in blood smears of a control subject (Figure 8B). In blood smears from young DS participants, abnormal RBCs and a classic dysmorphology (bi-lobed neutrophils) were frequently observed with very granular hepcidin in the RBC membranes (Figure 8C). In contrast, normal neutrophils positive for hepcidin (green) and platelet marker CD42b (red) were found in control smears (Figure 8D). The blood smear from a young DS participant (DS12, 30 years old) showed more hepcidin protein expression within MNC and around the neutrophils, and a very limited number of CD42b-positive platelets was seen (Figure 8E). One older DS participant (DS28, 65 years old) showed very abnormally clumped RBCs and low platelet counts (Figure 8F), whereas another young DS participant (DS19, 32 years old) displayed soluble hepcidin around the MNCs, and Aβ42 was found scattered in the platelets (Figure 8G). Two young DS participants’ blood sample smears (DS18, aged 32 years and DS55, aged 39 years, with the *TREM2 R47H*, C to T mutation), that showed abnormally shaped RBCs (Figure 8C) with abnormal accumulation of TREM2 around the MNC (Figure 8I), were reported previously [29]. Another blood smear from a control participant when stained with glycophorin and TREM2 showed TREM2 being present around the RBC membrane (Figure 8H). To confirm that hepcidin and RUNX1 are expressed in human bone marrow megakaryocytes, two DS bone marrow slides were stained with RUNX1, hepcidin and FPN. Both proteins were expressed in the bone marrow blast cells and co-localised where FPN was expressed in RBCs (Figure 8J,K, Table 3). These findings indicate that RUNX1 protein is crucial for processes such as the self-renewal of haematopoietic stem cells and differentiation of the myeloid and lymphoid cell lineages, whereas hepcidin may be essential for normal development of RBCs and platelets, as well as transporting soluble proteins. TREM2 mutation causes Aβ42 accumulation in the erythrocytes, as we reported previously [29].

## 4. Discussion

Down syndrome (DS) is characterised by progressive age-related cognitive and functional impairment associated with memory loss and premature ageing [3,28,29]. There are many factors involved in the neuropathology of AD dementia (ADD) in DS that contribute significantly towards the pathophysiology and enhanced ageing processes. The mechanism behind the early onset of AD pathology is not fully understood, but in this paper, we report that the overexpression of certain genes on chromosome 21, particularly in the Down syndrome critical region 1 (DSCR1) that includes *APP*, *Cu/Zn superoxide dismutase* (SOD1), S100β and *RUNX1* transcription factors, play crucial roles [2]. It has been hypothesised that changes in gene expression in chromosome 21 are likely to affect the expression of genes on other chromosomes through the modulation of transcription factors, immune regulation and other factors [7,11,17].

This study was aimed at determining how the overexpression of genes in triosomy-21 (i.e., *APP*, *S100β*, *SOD1* and *RUNX1*) and other disomic genes affects (possibly via amyloid protein transport and clearance) neuroinflammation and accelerates ageing. Although dysfunction of APP processing is believed to be the key upstream factor in the pathogenesis of AD [55], neuroinflammation and activation of innate immunity are considered early events in the genesis of AD and in DS dementia [72]. In DS and in LOAD, neuroinflammation has been linked to the exacerbation of both SP and NFT pathology, as well as the clearance of Aβ from amyloid plaques [56,57,58]. We therefore characterised the senile plaque (SP) formation and glial cell defects in DS. Plaque pathology was quite different between the young and old DS brains. In a younger (42 years old) DS brain, abundant cotton wool appearance of SPs, composed of Aβ42, was seen throughout the cortex (Figure 1A). Most of the plaques were formed from a cluster of small swollen neurites surrounded by neuropil without an amyloid core, whereas in the periphery, some other mature SPs were surrounded by a halo of Aβ42 (Figure 1A). The SPs were composed of GFAP-positive astrocytes, and the end-feet of astrocytes surrounded the endothelial layer of blood vessels. It is likely that end-feet of astrocytes could import nutrients to brain parenchyma and export lipid and waste products to the blood vessels [73]. These findings support the notion that not only neurons but also astrocytes contribute a great deal in amyloid clearance and plaque formation.

There are many genes triplicated on Hsa21, particularly Cu/Zn superoxide dismutase (SOD1) and S100β, which are associated with oxidative stress, and their overexpression may contribute to a genetic imbalance in DS that potentiates the development of ROS [12,35]. S100β is a member of the EF-hand type of calcium binding S100 protein family found in the brain and triplicated in the DSCR1 region [74]. Under normal physiological states, S100β is expressed predominantly in the astroglia, and also to a lesser extent in the neurons, microglia and oligodendrocytes [69]. S100β promotes neurotransmitter development and maturation in the mammalian brain, and it has a role in the cascade of glial changes associated with neuroinflammation [69]. We analysed DS and age-matched controls with the S100β antibody by IF. In control brain, S100β-positive normal astrocytes with thin filamentous processes of end-feet were present and surrounding the blood vessels (Figure 2J–L). Whereas, in DS brain, when S100β-positive, the end-feet of astrocytes were damaged, disorganised and aggregated around the wall of blood vessels, indicating changes to the cytoskeleton and altered astroglial morphology (Figure 2M–O). Within astroglial cells, S100β regulates Ca^2+^ levels, and once activated by Ca^2+^, S100β interacts with intermediate filaments including GFAP and vimentin and might have a role in ageing and neuropathology of DS [74]. It was also described that overexpression of human S100β exacerbates cerebral amyloidosis and gliosis in the mouse model of Alzheimer’s disease [75]. Our findings support that S100β might be involved in amyloid clearance from astrocytes, and due to the failure of transport, may lead to SP formation. This could be one factor of premature signs of ageing and reactive gliosis in DS.

Another protein that is triplicated in the DSCR1 region is superoxide dismutase 1 (SOD1). To cope with this natural source of ROS, it is equipped with a sophisticated antioxidant defence system which counteracts and controls ROS levels to maintain physiological homeostasis [76]. Superoxide dismutases (SODs) are a class of enzymes that catalyse the conversion of superoxide radicals to oxygen and hydrogen peroxide. SOD activity relies on a specific catalytic metal ion, which could be manganese (MnSOD), iron (FeSOD), nickel (NiSOD) or copper (Cu/ZnSOD) [77]. It is commonly known for its scavenging activity of ROS, but recent work has uncovered additional roles in modulating metabolism, maintaining redox balance and regulating activation of nuclear gene transcription following exposure to oxidative stress [78]. The impairment of SOD1 antioxidant function or the toxic gain of function have been associated with diseases, including DS [79]. We have seen a significant upregulation of SOD1 in the DS serum and high SOD1 staining, particularly in the neurons in senile plaque (Figure 4O). This observation requires further evaluation in larger samples to ascertain the role of SOD1 in DS.

Iron proteins (ferritin, hepcidin, FPN and DMT1) are involved in inflammation [38,41,48,55]. Hepcidin is a key regulatory protein of iron homeostasis, and due to its size (2.8 kDA), it could easily transport iron from periphery to the brain parenchyma [38]. In DS brain, hepcidin was present in the activated astrocytes around the SP and in the endothelial cells of the blood vessels, indicating that hepcidin is involved in glial iron transport from astrocytes to the blood vessels [38]. We reported that hepcidin is present in the end-feet of astrocytes that surround the blood vessels and is involved in iron transport [39]. Additionally, we reported that the choroid plexus (CP) is damaged in DS and AD brains and CP is a conduit between the peripheral circulation and central nervous system via the cerebrospinal fluid [66]. Here, we showed that, in mouse brain, DMT1 is expressed in the monolayer of epithelial cells of CP on luminal sites, whereas FPN was found in the abluminal site of CP as well as in the macrophages very close to the epithelial cells of CP. Similarly, in human brain, hepcidin was visible in small vesicles in the epithelial cells of CP. These results support our previous findings of hepcidin being transported through exosomes or other small vesicles, while FPN is transported via macrophages to the brain parenchyma [39].

Hepcidin and ferroportin are expressed in the oligodendrocytes of WMT. We previously reported that WMT defects were seen in DS brain white matter (WM), and, in the rat brain, hepcidin and FPN expressed in the WMT of the corpus callosum (CC) and were involved in myelination [53,66]. Hepcidin and FPN could be involved in glial activities not only in astrocytes but also in the oligodendrocytes to transport amyloid and other proteins [38]. To assess this aspect, we extended our investigation to evaluating hepcidin and FPN in WM in controls and DS brains. Hepcidin staining was visible in the WM, with normal appearance in the control brain. Contrastingly, very granular appearance was noticed in the cell bodies of oligodendrocytes in the DS brain. Hepcidin was also present in the WMT of basal ganglia (BG), with punctate accumulation of vesicular hepcidin in the cell bodies and around the blood vessels. It was reported that FPN could bind to APP in AD brain [54]. Here, we could show that both FPN and hepcidin have important roles in transporting amyloid and iron via WMT.

OLIG1 and OLIG2 are oligodendrocyte proteins that triplicated in the DSCR 1 region in DS. Both these genes regulate multiple steps in the formation of neurons and astroglia within the CNS [80,81]. These proteins also promote the formation and maturation of oligodendrocytes within the brain [70]. We investigated OLIG2 and myelin basic protein (MBP) expression in control and DS white matter of the cortex and in the basal ganglia (BG). In the control brain, MBP expresses in the myelinated sheath of oligodendrocytes, and OLIG2 localised in the cell bodies, whereas in the DS brain, myelin sheaths were shrunken and damaged. These findings indicated that OLIG2 gene defect is also implicated in myelination defects in DS, and this impairment in white matter could be one of the factors for cortico-basal degeneration in DS.

Abnormalities in the immune system are also seen in people with DS and can play a role in the pathogenesis of AD. Besides iron levels in the circulation, inflammation is yet another factor regulating hepcidin transcription in the hepatocytes. Inflammatory cytokines, IL-6 and IL-1β, are the main pathological mediators causing cytokine release [39,61]. Our serum analysis showed hepcidin and IL-6 levels to be significantly higher in the old DS participants but not in the young DS participants (*p* < 0.0001). Additionally, in the cortex of one participant with DS, hepcidin was present in damaged neurons with a halo around the nucleus, and IL-6 was visible in the microglia, suggestive of signs of nuclear damage and cell death with disease progression (Figure 5N). These findings indicate that both proteins, hepcidin and IL-6, are involved in host defence mechanisms during neuroinflammation, and failure results in cell death.

The identification of a novel variant in the gene TREM2 has refocused attention on inflammation as a major contributing factor in AD. Genome-wide association studies have shown that a rare mutation of *TREM2* (R47H, nucleotide C to T substitution) correlates with a heightened risk of developing AD [63,64,82]. In vitro studies indicate that TREM2 deficiency reduces the efficacy of amyloid clearance [83] and thus can contribute to the pathogenesis of AD [63,65]. In our cohort, two young DS participant’s blood smears (DS18, aged 32 years, and DS55, aged 39 years, with the *TREM2 R47H* mutation) showed abnormally shaped RBCs in their smears with abnormal accumulation of TREM2 around the MNC (Figure 8C,I) [29]. Both DS subjects developed dementia within five years. We demonstrated impaired trafficking of TREM2 to the plasma membrane of erythrocytes in DS, which could affect amyloid clearance from blood cells [29].

RUNX1 protein is crucial in processes such as self-renewal of haematopoietic stem cells and differentiation of the myeloid and lymphoid cell lineages [24]. We analysed RUNX1 protein in DS and control blood smears that were co-labelled with hepcidin and an erythrocyte plasma membrane marker, glycophorin. Both proteins were found to be co-localised in the RBCs membranes, suggesting RUNX1 expression on the surface of RBCs (Figure 8A). Hepcidin was observed to be scattered and present extra-corporally to RBCs and within the cell bodies of mononucleated cells (MNC) in the blood smear of a control subject. In contrast, in blood smears from young DS participants, classic dysmorphology (bi-lobed neutrophils) with significantly higher hepcidin around the MNC was frequently observed. In control blood smears, hepcidin was present in neutrophils and platelets, whereas in DS, a very limited number of CD42b-positive platelets were seen. For two DS bone marrow slides, when stained for RUNX1 and hepcidin, both proteins expressed in bone marrow blast cells and co-localised. These findings indicated that RUNX1 protein is crucial for processes such as self-renewal of haematopoietic stem cells and differentiation of the myeloid and lymphoid cell lineages, and that hepcidin may be essential for normal development of RBCs and for transporting soluble proteins. RUNX1 and hepcidin were observed in erythro-megakaryocytes and in platelets, suggesting a myeloid origin, and impairments of these proteins could cause myelodysplastic syndromes (MDS) and premature ageing of erythroid cells [29].

Hepcidin expression has been studied in many neurological disorders, including multiple sclerosis (MS), stroke, Parkinson’s disease (PD) and AD [38,84,85,86]. In a preliminary study of acute ischemic stroke (AIS), hepcidin levels were analysed and it was reported that plasma hepcidin levels were significantly higher in ischemic stroke patients compared with the control groups [84]. It was also mentioned that the hepcidin level significantly increased one week after diagnosis and after treatment with thrombolysis. This was an informative paper, but the authors did not explain why the hepcidin level increased or its importance in AIS [84]. Another group analysed serum hepcidin levels in multiple sclerosis (MS) in relapsing and progressive patients (active vs. inactive subjects, *n* = 71) and compared with healthy controls (*n* = 16) [85]. Serum hepcidin levels were significantly higher in MS participants (26.9 ng/mL) compared to (17.3 ng/mL) in healthy controls. The authors concluded that after correction for age and sex, the hepcidin difference was insignificant [85]. There are many unexplained questions in this study, and one of the important points is that hepcidin levels increased in the very early stages of inflammation, and it could be very different in MS (relapsing vs. progressive: active versus inactive stage) subjects. The disease duration was loo long, and that may affect the serum hepcidin production as well as the requirement to analyse a larger control group. The level of hepcidin in inflammatory diseases such as MS is very important and requires further research. Hepcidin protein may increase in early disease processes in other neurological disorders. Recently, another article reported higher serum pro-hepcidin levels in Parkinson’s disease (PD) after treatment of deep brain stimulation [86]. Hepcidin has immunomodulatory and neuroprotective effects, as we have seen in another ALS rat model, where after injury, hepcidin levels increased in new neurons, suggesting that hepcidin might increase in neurogenesis [87].

Our results strongly signify that DS pathology is multifactorial, and epigenetics and gene triplications influence many genes that not only cause neuronal but also extensive glial defects involving astrocytes and oligodendrocytes. These glial cells are essential for clearance of amyloid protein, and failure of such processes can lead to inflammation and cell death, as is observed in DS.

## 5. Conclusions

In Down syndrome, Aβ deposits begin to accumulate as early as in childhood and increase progressively with age. This is primarily due to the triplication of the DSCR1 region on chromosome 21, where *APP*, *S100β*, *SOD1*, *OLIG2* and *RUNX1* genes are located. RUNX family transcription factors play critical roles in the development of neural and non-neural cells. RUNX1 protein is also crucial for processes such as self-renewal of haematopoietic stem cells and the differentiation of myeloid and lymphoid cell lineages. Hepcidin, however, may be essential for normal development of RBCs and for transporting soluble proteins. RUNX1 and hepcidin were observed in erythro-megakaryocytes and in platelets, suggesting a myeloid origin of these proteins. SOD1 and S100β, on the other hand, have been associated with oxidative stress and ROS formation that could potentially lead to astrogliosis and neuroinflammation in DS.

Various immunological and haematological abnormalities are developmental and are common in DS, suggesting that inflammation and activation of innate immunity are early events in DS that may play a role in the pathogenesis of AD. The iron regulatory genes, ferritin, FPN and hepcidin, are involved in glial iron transport from astrocytes to blood vessels, and impairment of this process also impacts neuroinflammation. Serum hepcidin and IL-6 could be involved in host defence mechanisms of neuroinflammation, as seen in PD [86]. The expression of TREM2 in myeloid cells, even in erythro-megakaryocyte progenitors, strongly supports an intrinsic immune factor involvement in ageing and abnormal cellular processes seen in DS. Soluble TREM2 protein carried by platelets and erythrocytes from the periphery enters the brain parenchyma via a subset of microglia/macrophages and/or exosomes. In addition, the TREM2 (R47H) mutation, which has been strongly linked with an increased risk of AD, was found in two DS participants, and in both, we observed gross morphological changes in megakaryocytes and erythrocytes, further supporting an aetiological link with neurodegeneration. The gene product of RUNX1 and hepcidin plays a vital role in haematopoiesis in DS. We propose that soluble TREM2, S100β and hepcidin, migrating from the periphery via the CP, modulate the blood–brain immune axis in DS and could form an important and hitherto neglected avenue for therapeutic intervention.

## Figures and Tables

**Figure 1 jcm-10-02909-f001:**
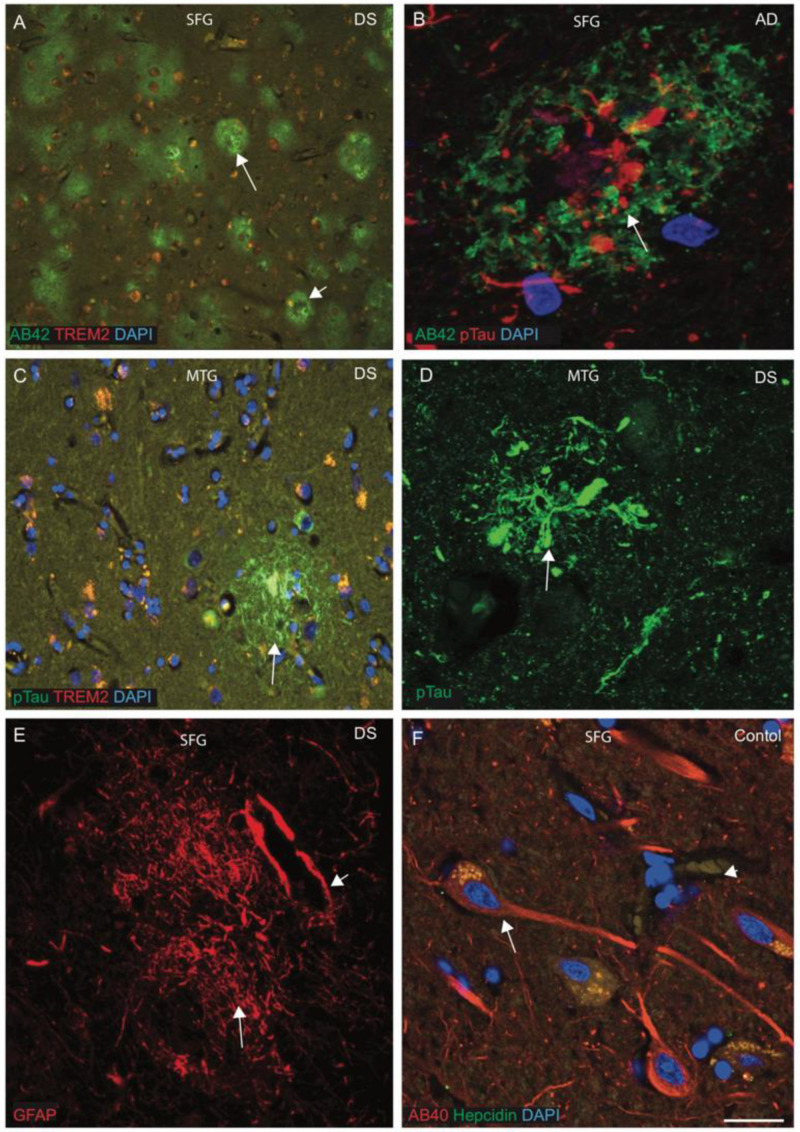
**Characterisation of senile plaques (SP) in DS brain.** To characterise the SP formation, the DS, AD and control brain sections from the superior frontal gyrus (SFG) and mid temporal gyrus (MTG) were labelled with double immunofluorescence using anti-Aβ42, phospho-tau (with AT8 antibody), anti-hepcidin and anti-TREM2, then counterstained with DAPI for nuclei (blue) and imaged with confocal microscopy. In DS brain, sections were stained with Aβ42 antibody, and an abundant cotton wool appearance of senile plaques (SP) was seen throughout the cortex (**A**, white arrows highlight the selected area of plaque formation). One AD brain section when stained with Aβ42 and pTau and analysed by confocal microscope showed a mature SP that contained many pTau-positive dystrophic neurites in the centre, with distinct Aβ42-positive astrocytes in the surrounding area (**B**). Another DS brain section from MTG was stained with pTau and TREM2, and neurofibrillary tangle (NFT) was visible in the core of the plaques, with some co-localisation with TREM2 in the SP and close by (**C**). In a confocal image of DS, MTG showed pTau-positive NFT visible very close to blood vessels (**D**). SFG samples from DS brain when stained with GFAP showed astrocytes surrounding the endothelial layer of blood vessels (**E**, short arrow highlights the blood vessel wall). One SFG from a normal brain section was stained with Aβ40 and hepcidin, and amyloid proteins were found to be evenly distributed in the neuronal cell body, axons and dendrites, whereas hepcidin was present in the cell bodies (**F**). Scale bar: (**A**,**C**) = 50 μm, (**B**–**F**) = 25 μm.

**Figure 2 jcm-10-02909-f002:**
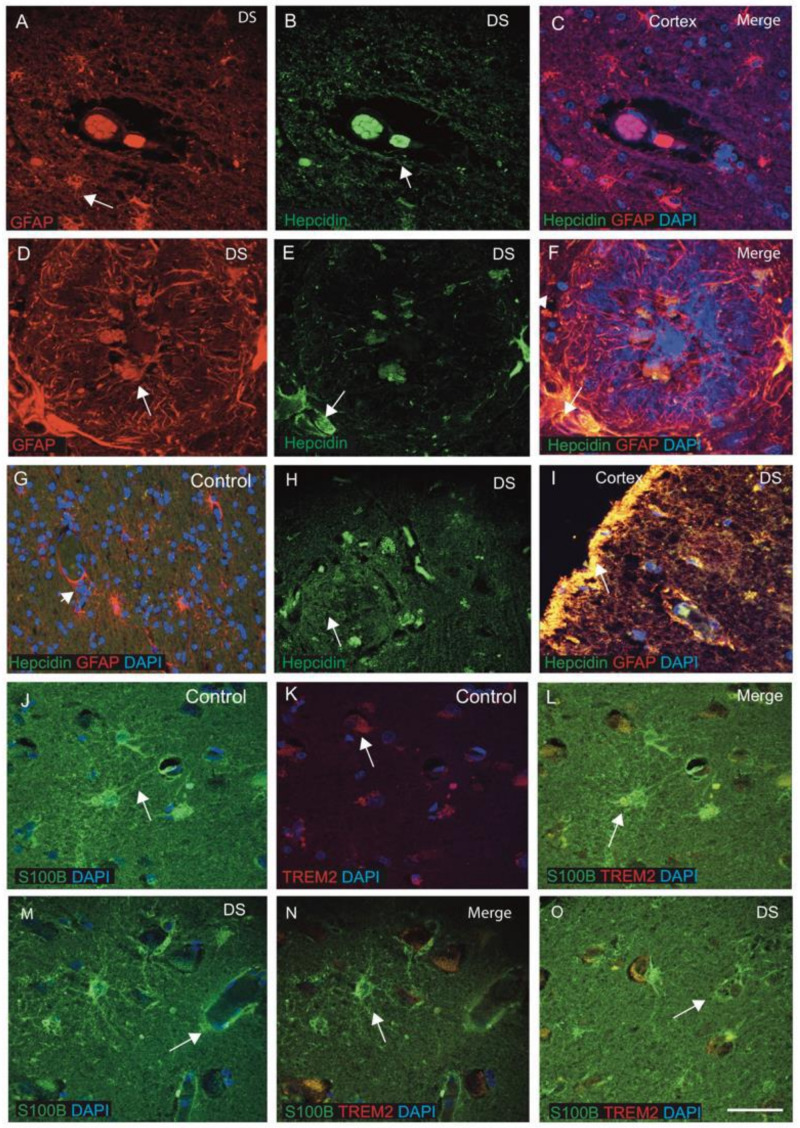
**Hepcidin and S100β-positive activated damaged astrocytes seen around the wall of blood vessels in DS.** DS and control brain sections from the cortex and close to the blood vessels labelled with double immunofluorescence (IF) using anti-hepcidin (green), anti-GFAP (red), counterstained with DAPI for nuclei (blue) and imaged with confocal microscopy. In DS brain sections, GFAP-positive activated astrocytes were present around the SP and surrounding the blood vessels (**A**–**C**, indicated with an arrow). Hepcidin expression was visible in the endothelial cells of blood vessels (**B**) and co-localised with the GFAP-positive astrocytes in the DS (**E**,**F**). In the senile plaques, GFAP-positive astrocytes were well-organised around the plaque formation (**D**–**F**). In the control brain section, normal astrocytes with fine processes were visible and co-localised with hepcidin in the cell bodies (**G**). A large number of small vesicles carrying hepcidin were present in the SP of the DS cortex and its surroundings (**H**). Very strong co-localisation of hepcidin with GFAP was noticed in layers I and II of the cortex (**I**). DS and control brain sections from SFG were stained with S100β and inflammatory protein TREM2. In control brain sections, thin filamentous processes of end-feet of normal astrocytes were present surrounding the blood vessels, positive for S100β and directly connected to the endothelial lining of blood vessels (**J**). TREM2 was present in the end-feet of astrocytes carrying soluble TREM2 (**K**) and co-localised in the cell bodies (**J**–**L**). In a DS brain section, S100β-positive activated astrocytes were found disorganised and aggregated around the blood vessel walls, indicating severe damage of endothelial cells of blood vessels (**M**–**O**). Scale bar: (**A**–**C**) = 25 μm, (**D**–**F**) = 15 μm, (**G**–**O**) = 30 μm.

**Figure 3 jcm-10-02909-f003:**
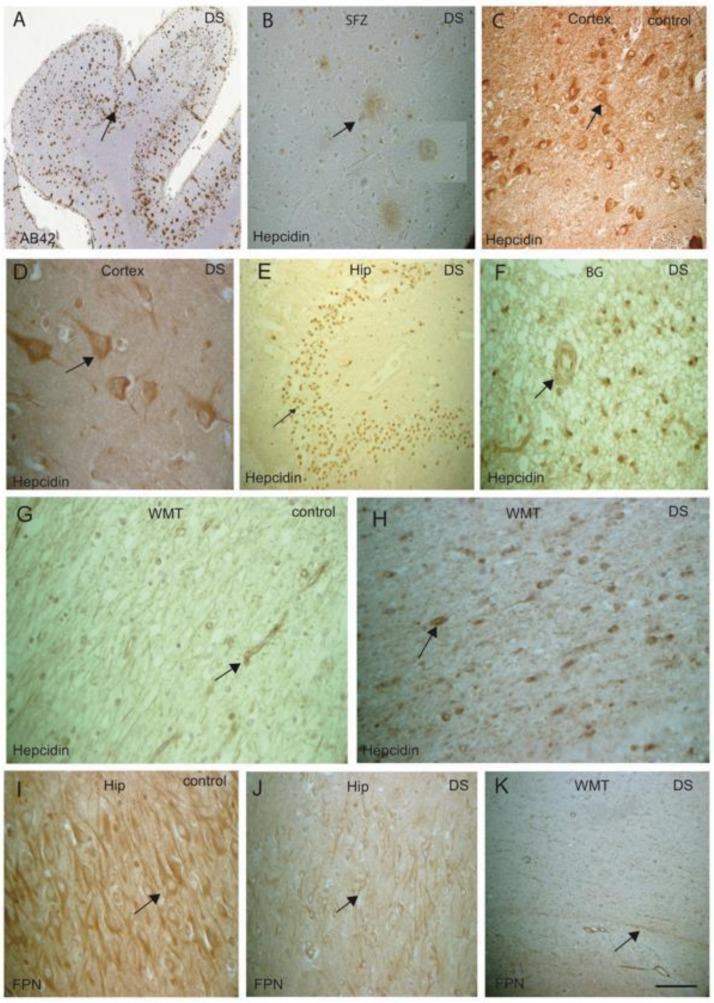
**Hepcidin and ferroportin expression in the oligodendrocytes of white matter tract.** DS brain sections from SFG stained with DAB, anti-Aβ42 and hepcidin antibody revealed strong signal of SPs, positive for Aβ42 and visible throughout the cortex (**A**, black arrows highlighting the selected area of plaque formation), hepcidin was visible in SPs (**B**). In control brains (cortex), abundant hepcidin was seen in the neurons and in the neuropils (**C**); in contrast, in the DS cortex, hepcidin staining was faint, present only in neurons and glial cells (**D**). In DS brain, hepcidin was present in the dentate gyrus (DG) granule cells (**E**), and in the basal ganglia (BG) as punctate accumulation of vesicular hepcidin in the cell bodies and around the blood vessels (**F**). Normal hepcidin staining was visible in the WMT, in the control brain (**G**), whereas very granular appearance was visible in the cell bodies of oligodendrocytes in the DS brain (**H**). Normal FPN expresses in the pyramidal neurons of HP in control brain (**I**), however, very faint staining was seen in the DS brain (**J**). Similarly, in DS brain, FPN expression was very faint and irregular in the WMT (**K**). Scale bar: (**A**,**E**,**G**–**H**) = 50 μm, (**B**–**D**) = 25 μm, (**F**,**I**–**K**) = 30 μm.

**Figure 4 jcm-10-02909-f004:**
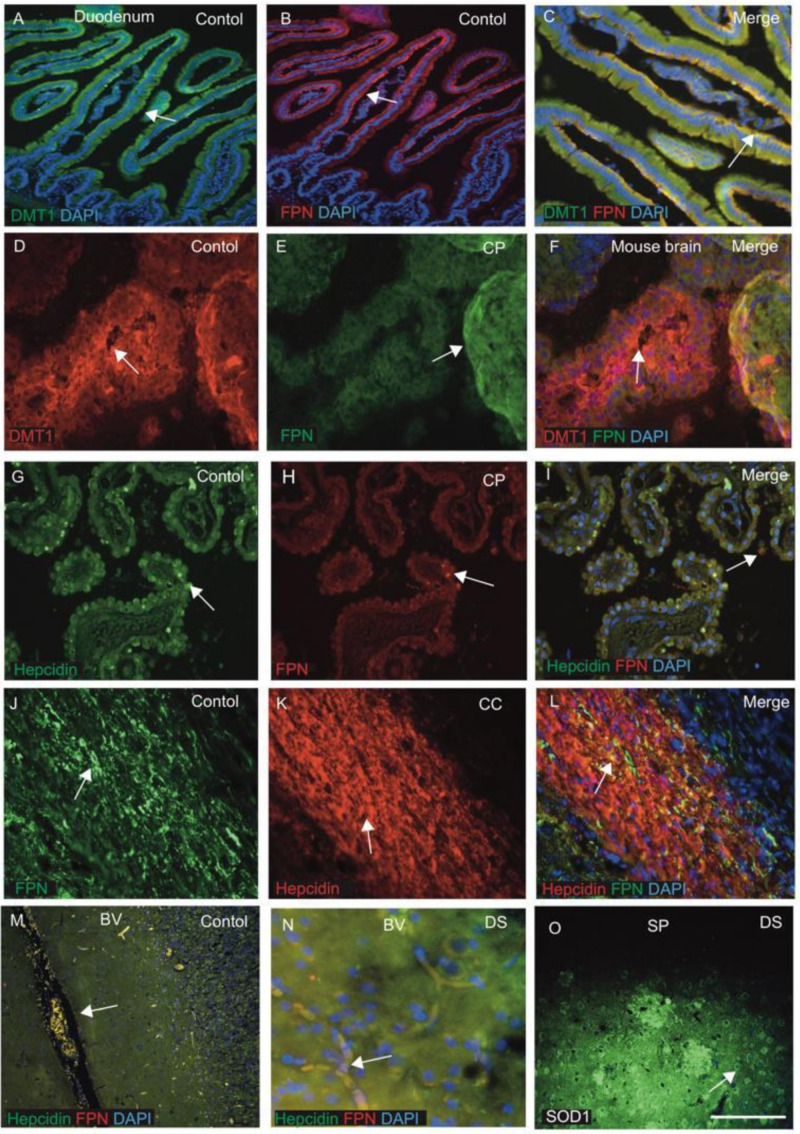
**DMT1, FPN and hepcidin regulate endothelial transport and SOD1 modulates redox balance in DS brain.** Tissue sections from mouse duodenum and choroid plexus (CP) were stained by IF using anti-DMT1 and anti-FPN antibodies and counterstained with DAPI for nuclei (blue). In duodenum, DMT1 staining was seen in the columnar cells within the crypts, particularly in the apical surface (**A**) and FPN at the basolateral site and in the central core of lamina propria containing blood vessels (**B**,**C**). The choroid plexus (CP) sections from mouse brain (**D**–**F**) stained with DMT1 were expressed in the monolayer of the epithelial cells of CP on luminal sites, whereas FPN was found in the abluminal site of CP, with some co-localisation observed (**D**–**F**). Similarly, for human CP when stained with hepcidin and FPN antibodies, hepcidin was visible in the epithelial cells of CP and FPN was seen in the macrophages very close to CP (**G**–**I**). Hepcidin and FPN are both visible in the oligodendrocytes located in the corpus callosum (CC), suggesting that iron transport may take place through CC (**J**–**L**). Hepcidin and FPN were visible in the blood vessels (**M**,**N**). A brain section from the cortex close to a SP was stained with SOD1 antibody, and very strong SOD1protein was seen in the plaque and in the peripheral neurons (**O**). Scale bar: (**A**–**F**) = 20 μm, (**G**–**L**) = 25 μm, (**M**) = 50 μm, (**N**,**O**) = 20 μm.

**Figure 5 jcm-10-02909-f005:**
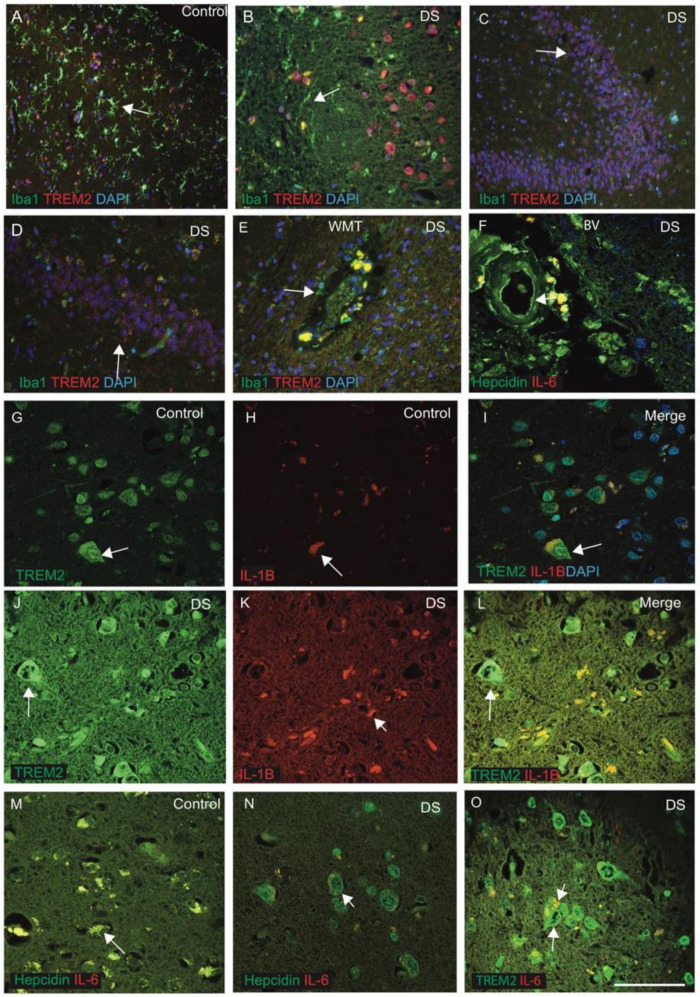
**Inflammatory changes in the DS brain were identified by TREM2, IL-1β and IL-6.** To investigate the inflammatory changes in DS, the brain sections from DS and control subjects were stained with microglia marker (Iba1) and TREM2. In control brains, Iba1-positive ramified microglia were visible in the cortex, whereas TREM2 was noticed in the neurons close to the blood vessels (**A**). In the DS brain sections, activated microglia was visible around the SP and limited co-localisation with TREM2 in the microglia (**B**). For sections from the HP and DG of DS brain when stained with Iba1 and TREM2, Iba1-positive activated microglia were present close to the neurons, whereas TREM2 expression was visible only in the damaged DG granule cells with limited co-localisation (**C**,**D**). The white matter (WM) close to the blood vessels (BV) in the DS brain showed large numbers of Iba1-positive microglia, close to the blood vessels, that might be involved in the clearing process of damaged oligodendrocytes, and TREM2-positive macrophages might be entering from blood vessels (**E**). For another DS brain section, close to the BV, when stained with hepcidin and IL-6, the staining patterns suggested that both proteins might be entering from the blood vessels (**F**). The brain sections from controls and DS subjects (from cortex) were stained with TREM2 and pro-inflammatory marker IL-1β. In control, TREM2 was visible in the neurons (**G**) and in other glial cells, and only very faint IL-1β expression was seen (**H**), with limited co-localisation in the cell bodies (**I**). In DS brain, TREM2 expression was seen in the neurons, whereas IL-1β was noticed in the microglia, close to the blood vessels and with some co-localisation noted (**J**–**L**). Similarly, for control and DS brain sections, when stained with hepcidin and IL-6, both the proteins were present in the cells close to the blood vessels and with some co-localisation (**M**), whereas in the DS brain, damaged neurons with a halo around the nucleus were positive for hepcidin (**N**), and IL-6 was visible in the microglia (**N**,**O**). Scale bar: (**A**–**C**) = 50 μm, (**D**–**F**) = 30 μm, (**G**–**O**) = 20 μm.

**Figure 6 jcm-10-02909-f006:**
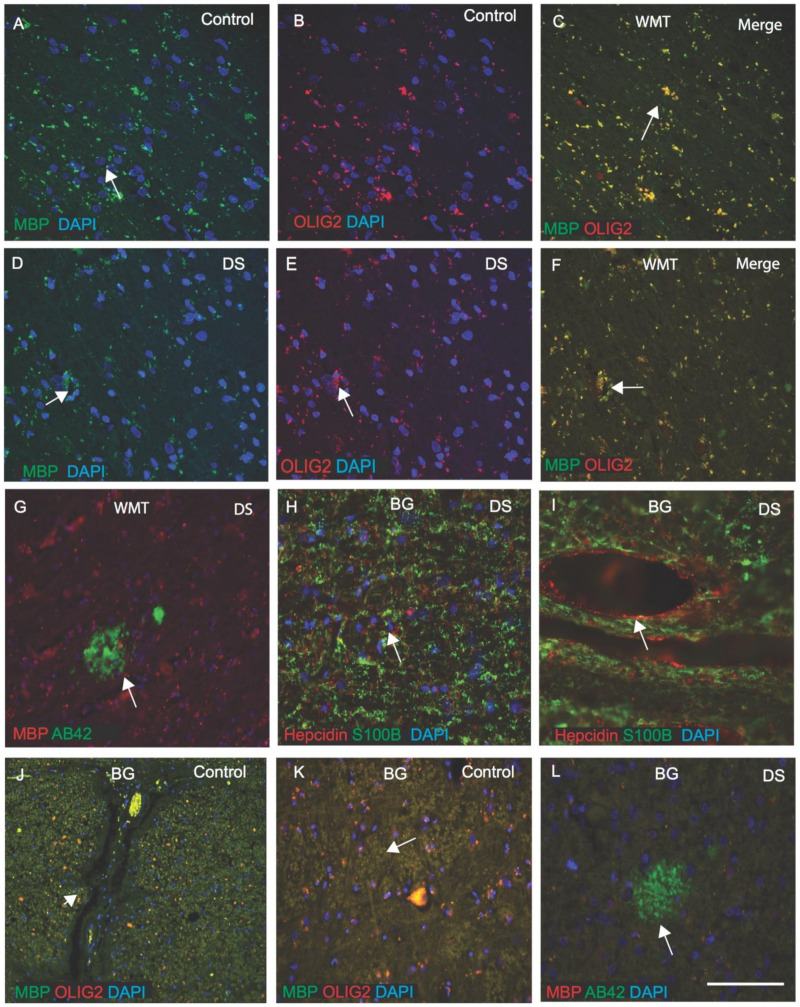
**OLIG2 defect impairs myelination in DS brain**. For DS and control white matter (WM) in the cortex and in the basal ganglia (BG), OLIG2 and myelin basic protein (MBP) expression were analysed by IF and counterstained with DAPI for nuclei (blue). In the control brain, MBP expression was seen in the myelinated sheath of oligodendrocytes, with OLIG2 localising in the cell bodies (**A**–**C**), and in the DS brain, myelin sheaths appeared to be shrunken and damaged (**D**–**F**). In DS brain, Aβ42-positive SPs were visible in the WM of the cortex (**G**). For another DS brain section from BG, when stained with hepcidin and S100β, both proteins were seen in the WM, and in particular, S100β was present in the oligodendrocytes (**I**), in astrocytes (**H**,**I**), whereas hepcidin was seen in the blood vessels walls (**H**,**I**). In control brain, particularly in the BG, both proteins, MBP and OLIG2, were seen in WMT (**J**,**K**), whereas, in DS brain, Aβ42-positive SPs were visible (**L**). Scale bar: (**A**–**F**) = 20 μm, (**G**–**L**) = 30 μm.

**Figure 7 jcm-10-02909-f007:**
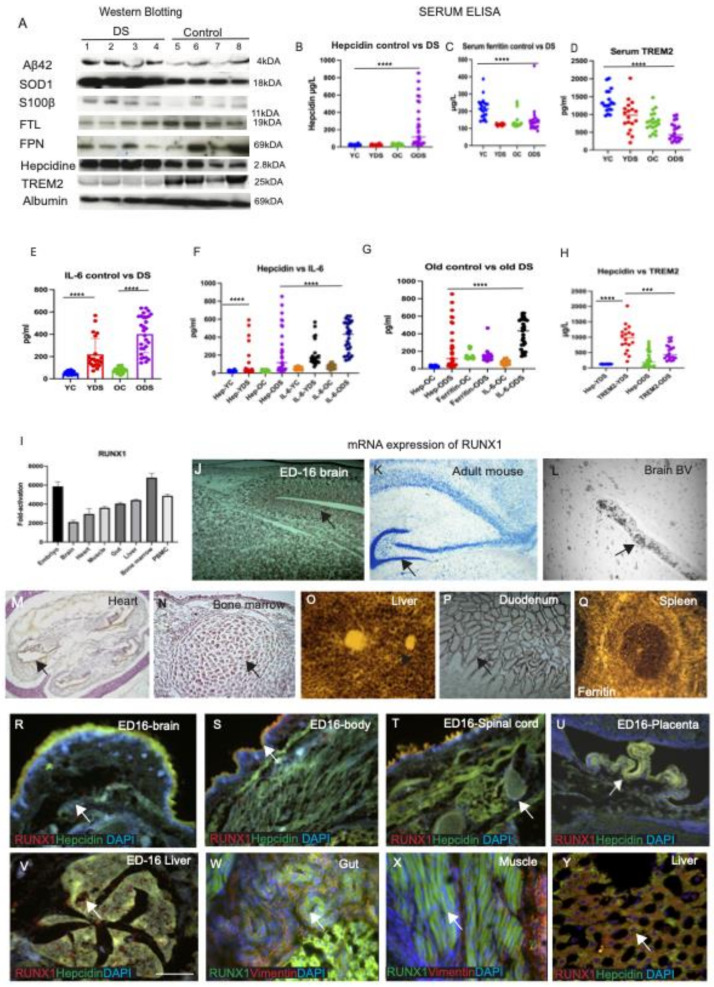
**Serum hepcidin and IL-6 could be involved in host defence****mechanism in DS brain.** RUNX1 mRNA expression was higher in the liver, bone marrow and duodenal endothelial cells, as assessed by in situ hybridisation. To investigate the effect of trisomy-21-related proteins (Aβ42, SOD1, S100β) and iron regulatory proteins (FTL, FPN, hepcidin), as well as inflammatory marker TREM2, on serum levels, a cohort of DS subjects (*n* = 18) and age-matched controls (*n* = 18) was analysed by Western blotting. Aβ42 (4 kDA) and S100β (11 kDA) levels were higher in the DS subjects compared to controls (Aβ42, *p* < 0.084, S100β *p* < 0.26, paired t-test, not significant) (Figure 7A and Appendix A). In contrast, SOD1 (18 kDA) level was found to be very high in DS serum (R^2^ = 0.64, *p* < 0.0007). However, FTL (19 kDA), FPN (69 kDA) and TREM2 (25 kDA) were significantly decreased in DS serum (*p* < 0.0001, *p* < 0.003 and *p* < 0.002, respectively) (Appendix A). Serum hepcidin was significantly higher in DS serum (*p* < 0.001) (**A**). Human serum from young and old DS (YDS, *n* = 23, and ODS, *n* = 24) and age-matched young and old control subjects (YC and OC, *n* = 25 in each group) were analysed by ELISA. Scattered plot showing the levels of serum hepcidin (**B**), ferritin (**C**), TREM2 (**D**) and IL-6 (**E**) (methods described in the main text and results were analysed by using one-way ANOVA that compared between groups, followed by Kruskal–Wallis or Bartlett’s test, corrected (Table 2), and a paired *t*-test, between YC vs. YDS and YC vs. OC, as shown in Appendix A). Serum ferritin was highest in controls (YC, 216.9 ± 171.1 μg/L, and OC, 211.43 ± 45.5μg/L) and lowest in the young DS (YDS, 122.56 ± 10.4 μg/L, compared to ODS, 139.83 ± 85.7 μg/L) (Figure 7C, R^2^ 0.31, *p* < 0.0001, Table 2). Serum TREM2 levels were the highest in young controls (YC, 1190.91 ± 150.2 pg/mL, and OC, 796.26 ± 147.65 pg/mL) and the lowest in the old DS (YDS, 894.30 ± 201.52 pg/mL, and ODS, 562.79 ± 112.75 pg/mL, R^2^ 0.48, *p* < 0.0001) (Figure 7D, Table 2). Serum IL-6 was higher in DS (YDS, 244.29 ± 144.4 μg/L, and ODS, 343.14 ± 320.1 μg/L) compared to both controls (YC, 52.62 ± 21.01 μg/L, and OC, 76.55 ± 31.4 μg/L, R^2^ 0.63, *p* < 0.0001) (**E**). Serum hepcidin and IL-6 levels were both highest in the old DS subjects (R^2^ 0.47, *p* < 0.0001 (**F**,**G**), Table 2). In contrast, serum TREM2 was higher in the controls compared to DS participants (R^2^ 0.77, *p* < 0.0001 (**H**), Table 2). Statistical differences were calculated by Mann–Whitney U test. *** *p* < 0.001 and **** *p* < 0.0001. The RUNX1 mRNA expression pattern by RT-PCR, in wild-type mouse (C57/bl6) tissues from different tissues, is shown in the **I**. This analysis revealed low levels of mRNA in the brain with a slightly higher signal observed in the heart, skeletal muscle, gut, liver and with the highest expression seen in the bone marrow and peripheral mononuclear leukocytes (PBMC) (**I**). To identify cellular levels, tissues sections from embryonic (ED16) and adult mice were analysed by in situ hybridisation (ISH). Very high mRNA was visible in ED16 mouse brain, throughout the cortex and in the brain cavity (**J**), but less in the adult brain, being only visible in the HP and DG (**K**), and clearly in the brain blood vessels (in the blood cells) (**L**). However, higher mRNA expression was seen in the bone marrow, endothelial layer of the heart and duodenum (**M**,**N**,**P**). The highest mRNA level was seen in the liver (**O**). Adult spleen sections were hybridised with ferritin as a control probe, and strong mRNA was visible (**Q**). We analysed by IF, using RUNX1 and hepcidin antibodies and counterstained with DAPI for nuclei (blue). Both proteins were observed in the embryonic brain, in the ED16 body and spinal cord (**R**–**T**), with co-localisation in the epithelial cells of the placenta (**U**). RUNX1 was distinctly visible in the embryonic liver (**V**), endothelial cells of the gut and in the muscles (**W**,**X**), and co-localised with hepcidin in the adult liver hepatocytes (**Y**). Scale bar: (**H**,**K**–**L**) = 30 μm, (**I**,**J**) = 50 μm (**M**–**O**) = 100 μm, (**P**–**Y**) = 25 μm.

**Figure 8 jcm-10-02909-f008:**
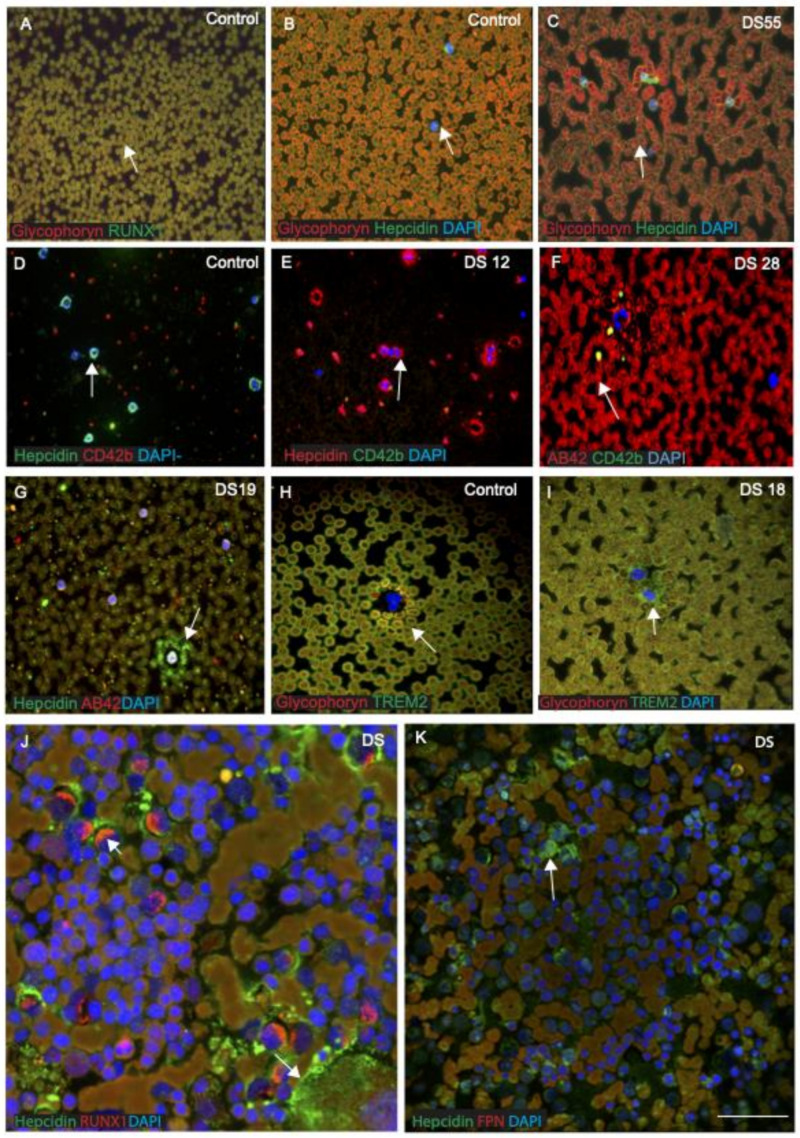
**RUNX1 and hepcidin were observed in the erythro-megakaryocytes and platelets, suggesting a myeloid origin.** We analysed RUNX1 and hepcidin expression in blood smears from DS subjects (*n* = 20) and age-matched controls (*n* = 20) and then imaged with confocal microscopy. DS and control blood smears were co-labelled with RUNX1 or hepcidin and glycophorin, an erythrocyte plasma membrane-marker. Both proteins (RUNX1 and glycophorin) were found to co-localise in the RBC membranes, suggesting that RUNX1 expressed on the surface of RBCs (**A**). Similarly, another smear from control and DS was stained with hepcidin and glycophorin. In control subjects, hepcidin was observed scattered, extra-corporally to RBCs and within the cell bodies of mononucleated cells (MNCs) (**B**). In young DS blood smears, a classic dysmorphology (bi-lobed neutrophils) with significantly higher hepcidin protein around the MNCs was frequently observed (**C**,**E**), whereas normal neutrophils positive for hepcidin (green) and platelet marker CD42b (red) were found in controls smears (**D**). The blood smear from a young DS subject (DS12) showed more hepcidin protein expression within MNC and around the neutrophils, and a limited number of CD42b-positive platelets was seen (**E**). One old DS subject (DS28) showed very abnormal clumped RBCs and low platelet counts (**F**), whereas another young DS (DS19) displayed soluble hepcidin around the MNCs, and Aβ42 was found scattered in the platelets (**G**). Two young DS participants’ blood sample smears (DS18, aged 32 years, and DS55, aged 39 years, with the *TREM2 R47H*, T mutation) showed abnormally shaped RBCs in their smears (**C**), with abnormal accumulation of TREM2 around the MNC (**I**), as reported in our previous publication [29]. A blood smear from a control participant was stained with glycophorin and TREM2, showing TREM2 being present around the RBC membrane (**H**). To confirm that hepcidin and RUNX1 are expressed in human bone marrow megakaryocytes, two DS bone marrow slides were stained with RUNX1, hepcidin and FPN. Both proteins were expressed in the bone marrow blast cells and co-localised where FPN expressed in RBCs (**J**,**K**). These findings indicate that RUNX1 protein is crucial for processes such as the self-renewal of haematopoietic stem cells and differentiation of the myeloid and lymphoid cell lineages, whereas hepcidin may be essential for normal development of RBCs and platelets, as well as transporting soluble proteins (Table 3). Scale bar: (**A**–**I**) = 20 μm, (**J**,**K**) = 10 μm.

**Table 1 jcm-10-02909-t001:** Characteristics of post-mortem (PM) brain samples of DS and age-matched control study population, showing age, gender, post-mortem time delay, Braak stages and cause of death.

Case Number	Category	Age	Gender	PM Delay (h)	Braak Stage	Cause of Death
Down Syndrome (DS)						
DS1	DS	56	F	6	6	Not known
DS2	DS	76	F	18	6	Septicaemia
DS3	DS	46	F	8	2	Not known
DS4	DS	52	M	24	6	bronchopneumonia
DS5	DS	64	M	31	6	Not known
DS6	DS	66	F	26	5	Not known
DS7	DS	67	M	8	5/6	Alzheimer’s disease (AD), Cerebrovascular disease (CVD)
DS8	DS	52	F	-	6	AD, Lewy body dementia
DS9	DS	52	M	-	5	Traumatic brain injury (TBI)
DS10	DS	76	F	18	6	Dementia
Controls						
C1	Normal	66	M	10.3	5	Cerebrovascular Disease/Dementia
C2	Normal	45	F	43.3	0	End stage renal failure/diabetic nephropathy
C3	Normal	54	F	10.3	0	Metastatic Myxoid Liposacroma/Bronchopneumonia
C4	Normal	52	F	30.3	1	Bronchogenic Cancer
C5	Normal	75	F	24	2	Cancer of the Ovary
C6	Normal	66	F	29.3	2	Metastatic Breast Cancer
C7	Normal	83	M	45	0	Not known
C8	Normal	68	M	48	0	Not known
C9	Normal	60	F	60	0	Not known
C10	Normal	66	M	74	0	Not known

**Table 2 jcm-10-02909-t002:** Characteristics of DS and age-matched control study population used to measure serum iron, ferritin, hepcidine, IL-6 and TREM2.

Characteristic	Young Controls (YC)	Young DS (YDS)	Old Controls (OC)	Old DS (ODS)	ANOVA Table	R^2^	*p*-Value	*p*-Value Summary
Male + Females	N = 25	N = 23	N = 25	N = 24	**F (DFn, DFd)**			
Age (years)	40 ± 10	30 ± 10	50 ± 27	41 ± 25				
**Iron Parameter**								
Serum iron (μmol/L)	25.07 ± 5.06	19.11 ± 4.5	20.18 ± 7.2	13.92 ± 7.4	32.47	0.4985	*p* < 0.0001	****
Serum Ferritin (μg/L)	216.9 ± 171.1	122.56 ± 10.4	211.43 ± 45.5	139.83 ± 85.7	13.64	0.3198	*p* < 0.0001	****
Serum Hepcidin (µg/L)	23.89 ± 19.1	376.36 ± 475.4	26.56 ± 16.1	241.02 ± 430.5	18.09	0.35	*p* < 0.0001	****
**Inflammatory Parameter**								
Serum IL-6 (µg/L)	52.62 ± 21.01	244.29 ± 144.4	76.55 ± 31.4	343.14 ± 320.1	55.78	0.6307	*p* < 0.0001	****
SerumTREM2 (pg/mL)	1190.91 ± 150.2	894.30 ± 201.52	796.26 ± 147.65	562.79 ± 112.75	22.22	0.4807	*p* < 0.0001	****

Data are expressed as number of patients, mean ± SD. Probability values (*p*) denote differences between control and DS groups. One-way ANOVA that compared between groups, followed by Kruskal–Wallis or Bartlett’s test (corrected), was used that compared to age between groups, followed by Mann–Whitney test. **** *p* < 0.0001.

**Table 3 jcm-10-02909-t003:** Characterisation of blood smears of Down syndrome and normal control subjects.

Cell Types	Controls *n* = 20		Phenotypes and Outcome	Down Syndrome *n* = 20		Phenotypes and Outcome
Blood smear	Young (YC), *n* = 10 (age 30–40 years)	Old (OC), *n* = 10 (age 41–65 years)		Young (YDS), *n* = 10 (age 30–40 years)	Old (ODS), *n* = 10 (age 41–65 years)	All older DS participants developed Alzheimer’s disease pathology.
Red cells (RBCs)	Normal (Normocytic)(Figure 8A).	Normal, often paler zone in the centre (Figure 8B).	Staining is more intense at the periphery.	RBCs closed and overlapping (Figure 8C).	Mildly microcytic (Figure 8F).	Microcytic and hypochromic blood film was seen in old DS subjects (Figure 8F)
Neutrophils	Normal	Normal	The cytoplasm is pale pink and contain fine violet granules.	Normal	Increased granulation of neutrophils.	The nucleus has 2–4 lobes of coarse chromatin. Occasionally seen multi-segmented.
Eosinophils	Normal	Normal	Two lobes to the nucleus	Normal	Normal	The cytoplasm was filled with large eosinophilic granules
Basophils	Normal	Normal	Similar size to neutrophils with nucleus.	Occasionally seen large purple/black granules (Figure 8G).	Normal	Found in two cases who had hypothyroidism.
Lymphocytes	Normal	Normal	Normal adult count.	Small lymphocytes.	The lymphocytes population is heterogeneous.	The nucleus was an irregular shape.
Monocytes	Normal	Normal	Large cells, the nucleus was often irregular or kidney shape (Figure 8D).	Lymphocytosis developed and atypical mononuclear cells was found (Figure 8F).	Cells have irregular nuclei, often with nucleoli (Figure 8G).	Some of the monocytes were activated T cells. (Figure 8C,F)
Platelets	Normal	Normal	Platelets were round or oval and had an irregular outline (Figure 8D).	Platelet count was very low (Figure 8E)	Platelet count was reduced, and cell fragmentation was visible (Figure 8F).	Low platelet count is common in young DS, who are prone to suffer from acute myeloid leukaemia (AML).
Bone marrow	Not analysed	Not analysed	Not analysed	The proerythroblast was seen, large cells with high nuclear to cytoplasmic ratio (Figure 8J).	Not analysed	Bone marrow (BM) was hypercellular with gross myeloid hyperplasia. The cytoplasm was dense and there was often a perinuclear pale zone (Figure 8K).
Protein	Expression in	blood	smears			
Glycophorin	Normal glycophorin-positive cell membrane (8A).	Normal glycophorin-positive cell membrane (Figure 8B).	Glycophorin is a siloglycoprotein of the membrane of a red blood cell.	RBCs were clumped but normal glycophorin expression (Figure 8C).	Glycophorin-positive membranes were clumped (Figure 8I).	Glycophorin protein could bind with ferritin and is involved in iron transport. DS participants suffer from hypoferritinaemia, that could damage RBC membranes.
RUNX1	Normal expression in the RBCs (Figure 8A).	Normal expression.	RUNX1 protein is crucial for processes such as the self-renewal of haematopoietic stem cells.	RUNX1 are expressed in human bone marrow megakaryocytes (Figure 8J).	There was marked dyshaemopoisis and modest increase of blast cells.	RUNX1 are expressed in bone marrow megakaryocytes, and differentiation of the myeloid and lymphoid cell lineages.
Hepcidin	Hepcidin-positive normal mononucleated cells (Figure 8D).	Hepcidin was found in the RBCs and in MNCs (Figure 8B).	Soluble hepcidin was seen in the RBCs and in MNCs (Figure 8B,D).	Hepcidin-positive monocytes, many of them were abnormal appearance (Figure 8E).	Soluble granular hepcidin was visible around the MNC and scattered throughout the smear (Figure 8G).	Hepcidin may be essential for normal development of RBCs and platelets, as well as transporting soluble proteins.
FPN	Normal appearance in RBCs and MNCs.	Normal appearance in RBCs and MNCs.	FPN expression was seen in all cell membranes.	Normal levels of FPN were present in RBC membranes.	Low levels of FPN were present in RBC membranes.	FPN proteins were expressed in the bone marrow blast cells.
TREM2	Normal membrane-bound TREM2 visible in RBCs membrane (Figure 8H).	Similar pattern but less intense was seen in OC.	TREM2 protein involved in erythrophagocytosis [29].	One young DS (DS18) carrying R47H mutation showed abnormally shaped RBCs (Figure 8I).	Another DS participant (DS55), carrying R47H mutation showed, and accumulation of soluble TREM2 around the MNCs.	TREM2 mutation impaired trafficking of TREM2 to the plasma membrane of erythrocytes in DS, which could affect amyloid clearance from blood cells [29].
Aβ42	Normal	Normal	Aβ42 accumulation was not seen in normal blood smears.	Aβ42 was visible in fractured platelets (Figure 8G).	Aβ42 was visible in in abnormal shape RBCs and co-localised in platelets (very low platelet count) (Figure 8F).	RBCs and platelets participate in amyloid clearance.
CD42b	Normal platelets (Figure 8D)	Normal platelets	CD42b is a platelet marker	Platelet count was very low (Figure 8E).	Platelet count was reduced, and cell fragmentation was visible (Figure 8F)	Low platelet count is also seen in acute myeloblastic leukaemia, which was seen in one young DS.

## Data Availability

All datas are described here in the paper.

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
