# Peer review of "Impaired Iron Homeostasis and Haematopoiesis Impacts Inflammation in the Ageing Process in Down Syndrome Dementia"

_jcm, 2021, doi:10.3390/jcm10132909_

Round 1
Reviewer 1 Report
A very well written publication. The experiment was properly planned and carried out. The results themselves are extremely detailed. Therefore, I believe that the publication should be published.
Below I present minor remarks:
1. introductuon is too long and may be slightly shortened,
2.the authors in the discussion should discuss the role of hepcidin in other diseases of the nervous system: stroke (doi: 10.1016 / j.jstrokecerebrovasdis.2015.03.031), multiple sclerosis (doi: 10.1177 / 2055217319885984), or PD (doi.org/10.1016/j newsletter. 2018.06.031).
Author Response
Response 1: We have shortened the introduction by deleting sections explain iron homeostasis. As this paper describe the role of different iron gene/proteins and other genes located in Triosomy 21 in Down’s syndrome. We felt it is necessary to describe these proteins in the introduction.
Response 2: In the discussion section, we have explained role of hepcidin in other neurological diseases including stroke, multiple sclerosis, Parkinson’s disease and Alzheimer’s disease (ref:38, 85-87) and outcome of those findings as suggested by reviewers. I would like thank to reviewer for these suggestions.
Reviewer 2 Report
Raha-Chowdhury et al. provide an assessment of several important proteins within groups of Down Syndrome (DS) patients and appropriate controls. They identify distinct morphologic differences in smears between 10 older DS patients (presumably all with AD or AD symptoms) and 10 age matched controls – all from post mortem samples. Additionally, they assess serum samples from both older and younger DS patients and also from older and younger controls. While the design of the study is quite good, the presentation of the results in a useful table format is lacking for many of the key proteins which are part of the stated main aim of the authors. Hopefully this can be readily improved. Additionally, the statistical methods reported in the current Table 2 are either not informative or not clearly specified. Again, hopefully this can be quickly altered by the authors. It would also be beneficial if the authors had some way of summarizing the characteristics identified in the smears across all 20 patients into an easily comparative table (rather than having to rely solely on text and image descriptions for only a subset of the 20 patients). Finally, it would be helpful for the authors to clearly state what is already known about these protein levels of similar subjects in other studies.
- The authors characteristically analyze smears of 20 patients which is a notable work. It would be helpful if an additional table was provided which summarizes the frequencies for both the 10 DS patients and for the 10 controls that the various morphologic features were identified (and hopefully broken down further by amount present – e.g., absent or rare, minimal, frequent … or some other classification).
- Table 2 provides helpful summaries of serum levels across YC, YDS, OC, and ODS for both iron parameters and for IL-6 and TREM2 – but omits including the levels for APP, S100Beta, SOD1, and RUNX1 (and also OLIG1 and OLIG2) … which are stated aims of the authors. Including these levels in Table 2 (rather than only having comparative results of these listed in the text) will be most helpful for readers and for ease of assessing claims of the authors. Particularly as these are a major part of the conclusions of the authors, an additional figure showing these results (in addition to their summary levels included in Table 2) would be appropriate.
- The statistical tests performed within Table 2 are either uninformative/incorrect or are mis-specified. A single p-value is provided for each row indicating an ANOVA across all 4 groups was likely performed. The Mann-Whitney test can only compare 2 groups, and if used, which two groups were compared is not specified. Furthermore, ANOVA is not really of interest here. Pairwise comparisons between YC and YDS, between OC and ODS, and between YDS and ODS should be performed with the corresponding 3 separate p-values for the Mann-Whitney test listed for each row. This is HIGHLY important to do, as a test across all 4 groups (e.g. using Kruskal-Wallis) could be significant merely because there is a difference between YC and OC and would give no insight into differences between controls and DS patients.
- The important proteins focused on in this paper have likely been assessed in other studies of AD and controls as well. The authors should comment on how their results compare to those in such studies (or emphatically state the novelty of this work). However, it would seem that genome-wide protein assessment has likely been done in similar cohorts and would contain potential for similarly answering some of these questions, albeit perhaps not the focus of those prior studies.
- The title of the manuscript uses language that is quite strong regarding the conclusions on hematopoietic impairment/iron defects/inflammation being causal in nature. These conclusions of causality are not justified by the results of this work and hence the title should be revised to reflect the associative nature of the findings within this study.
Other points:
* Section 3.7 (lines 580-589) shares some significant results; however, the results in the manuscript text refers only to significance levels and not effect sizes. Either a fold-change or the mean/median levels within each group should also be stated.
* Line 330 the authors claim that a paired Student t test was used – but it does not appear that a paired test should be used anywhere in the manuscript – can the authors please clearly indicate where a paired test was used if appropriate? As mentioned earlier, it is likely beneficial that a Mann-Whitney test be the sole test used throughout the entire paper – but please provide justification if any other tests are thought to be appropriate.
* Figures 7B-7G should be made clearer so that the x- and y-axis labels can be clearly read. Would help if sample sizes were also included below the x-axis labels (or in the figure legend, preferable not just in the text).
*line 625 does not match with the sample sizes in table 2 --- was it likely n=25 for YC and n=25 for OC, and not n=50 for YC and n=50 for OC?
*There appears to be a significant difference in the PM delay between the DS and controls – if so, this should likely be mentioned
*Figures of the smears could be rearranged so that DS and control samples are grouped separately rather than interspersed – though perhaps the authors felt the flow of their description is benefited by the present order – this point is merely mentioned for consideration.
*Please provide the corresponding transcript and ideally the codon/position with human genome version for the TREM2 R47H mutation listed on line 706.
*Perhaps this is a technical journal upload issue - the supplementary data are merely the main figures zipped together. No supp. material are referenced in the paper.
*The article reads extremely well. However, there are many typos throughout which should be corrected, e.g. Abstract "present of RUNX1" should be "presence of RUNX1", Abstract: rephrase "and hepcidin are plays a critical role", "while matter tract (WMT)".
Author Response
Response 1: As reviewer suggested, in current revised manuscript we have added an additional table (Table 3) representing figure 8. We have characterised all different types of blood cells present in DS (n=20) and in controls (n=20). Additionally, we have explained protein expressions pattern of Aβ42, hepcidin, glycophorin, TREM2 and CD42b in different cell types.
Response 2: In initial experiment (Figure 7 A), we have analysed serum levels for Aβ42, S100β, SOD1, hepcidin, FPN and TREM2 by Western blotting (WB). Due to a different batch of protein samples preparation and loading showed heterogeneity in different batches of WB analysis. These were semi-quantitive analysis, and we measured fold changes and calculated using paired student t test as shown in figure S1. RUNX1 was not informative in WB analysis, therefore we have analysed RNA levels by RT-PCR and in-situ hybridisation, followed by immunofluorescence (IF).
Response 3: Statistical analysis of ELISA data (Table 2) is now divided in two tables. The comparison between two groups (YC versus YDS and OC versus ODS) was analysed by paired t test as shown in figure S1. Serum samples by ELISA was evaluated (in four groups) by nonparametric statistical analysis (Kruskal-wallis or Bartlett’s test, corrected and post hoc with Mann-Whitney U test and showed in Table 2.
Response 4: The main focus of this paper is to determine expression of iron regulatory proteins (hepcidin, ferroportin and Divalent metal protein I (DMT1) in central nervous system (brain) and in serum of Down’s syndrome participants and age matched controls. Hepcidin, ferroportin and DMTI were well characterised in the liver, duodenum and serum in iron deficient and iron overloaded subjects (ref: 40-51). We previously, reported Hepcidin and ferroportin’s role in Alzheimer’s disease (ref:38), whereas, comparison of these proteins in DS brain and then its co-relation with triosomic proteins (RUNX1, S100β, SOD1, OLIG1 and OLIG2) are novel. These triosomic proteins were described in other publications as referred (ref:13, 21, 70, 71), but not compared with iron homeostasis proteins. In the discussion section, we have explained role of hepcidin in other neurological diseases including stroke, multiple sclerosis, Parkinson’s disease and Alzheimer’s disease (ref:38, 85-87) and outcome of those findings.
Genome-wide protein assessment was reported on AD, particularly a large number of publications reported on TREM2 (Ref: 65,66). We have first time reported the TREM2 mutation in DS population (Ref: 29, 39)
Response 5: In this manuscript we described iron homeostasis and neuroinflammation in DS. To in-line with the findings, we change the title to “Impaired iron homeostasis and haematopoiesis impacts inflammation in the ageing process in Down’s syndrome dementia”.
Response 6: Serum samples from DS and controls analysed by WB and ELISA. Comparison of YC versus YDS and OC versus ODS was analysed by a paired Student t test (Shown in Supplementary S1). Serum samples by ELISA was evaluated (in four groups) by nonparametric statistical analysis (Kruskal-wallis or Bartlett’s test, corrected and post hoc with Mann-Whitney U test and showed in Table 2.
Response 7: Figures 7B-7H, x axis showed different group of samples and y-axis showing protein concentration, font sizes increased from 8 to 12. The numbers of samples in each group is mentioned in the table 2, n=25 for YC and n=25 for OC, and n=23 for YDS and n=24 for ODS (see Table 2 for details).
Response 8: PM delay between the DS and controls were checked by co-author (neuropathologist), but none of the samples had any deteriations. We have used these samples for several other studies and published before (ref:29, 39, 67).
Response 9: TREM2 variant rs75932628, substitution of T to C, changes (R47H mutation) significantly associated with Alzheimer’s disease and Down’s syndrome (ref 64, 65, 67, 83).
Response 10: We have also made improvement to the language throughout the manuscript for ease to understanding without altering any factual content and corrected all spelling mistake as far as we could see. We would like thank to reviewer for these suggestions.
Round 2
Reviewer 2 Report
The authors have considerably improved the manuscript addressing the points that had been previously raised. In particular the addition of Table 3 and the added paragraph in the Discussion summarizing previous findings with regards to hepcidin expression were extremely helpful - thank you.
Also appreciated is the addition of Table S1 which now contains group by group comparisons. Corrections made to the sample sizes that were previously listed now explain why a paired t-test was used in many comparisons. The authors corrected many typos and it is much improved. Some typos still remain which should hopefully be caught by the copyeditors or authors at the proof stage.
Good job
Line 630: "between YC vs YDS and YC vs OC shown in (Table S1)" should probably be "between YC vs YDS and OC vs ODS shown in (Table S1)"